# Beyond the Stability-Exploration Dilemma: Environmental Regularization for LLM Policy Optimization

## Abstract

Policy optimization (PO) has advanced Large Language Models (LLMs), yet training remains constrained by a stability–exploration trade-off. We analyze the coupling between the input environment and the policy in LLM RL, and *decouple* parameter regularization from the optimization objective by moving regularization to the input side. Concretely, we propose **Environment-Regularized Policy Optimization (ERPO)**, instantiated with **Query-KL (QKL)**, which penalizes the KL divergence between the evolving *query* distribution and a fixed reference. By regularizing the input (query) distribution rather than the action (response) distribution, QKL indirectly controls policy drift induced by environmental shift while preserving exploration. To avoid premature convergence, we introduce a *query-weighted advantage* that reweights updates according to estimated query prevalence, reducing estimator variance and improving robustness. Across diverse mathematical reasoning benchmarks, ERPO achieves KL control comparable to methods with explicit policy regularization, while delivering stronger final performance and smoother training dynamics. Temperature-swept sampling further indicates more stable long-horizon behavior. These results suggest that making the input environment a first-class object—via QKL and query-weighted advantage— is a principled and practical route to improve the stability–exploration trade-off in PO for LLMs.

## 1 Introduction

**Background and challenge.** Policy optimization (PO) methods have become the de facto recipe for post-training large language models (LLMs), spanning trust-region style updates (TRPO/PPO) and preference-based objectives (DPO) together with broader RLHF/RLAIF variants (Schulman et al., 2015b; 2017; Ouyang et al., 2022; Bai et al., 2022; Rafailov et al., 2023). Despite impressive progress in mathematical reasoning and beyond, practitioners still face a persistent dilemma: *how to trade off training stability against effective exploration*. In long-horizon runs, optimization noise and distribution shift tend to accumulate, leading to oscillations and occasional collapses.

**Instability from the input side.** We argue that a key—and under-controlled—source of instability is *environment non-stationarity* induced by the **query distribution**. During RL fine-tuning, the inputs used for training are sampled from a mechanism that *co-evolves* with the policy (e.g., active data selection, prompt generators, curriculum schedulers). As the policy changes, the conditional likelihood of future prompts also shifts, altering the effective training environment and amplifying gradient variance. This mirrors classic RL settings in which either the initial-state distribution or the transition kernel drifts over time; non-stationary and robust RL therefore advocate explicit distributional control (Padakandla, 2021; Iyengar, 2005; Nilim & El Ghaoui, 2005). A related lesson from imitation learning is that policy updates induce covariate (state) shift, motivating interactive data aggregation such as DAgger/AggreVaTe (Ross et al., 2011; Ross & Bagnell, 2014).

**Limitations of action-only regularization.** Recent LLM work has started to surface the role of prompt distributions. EVA frames open-ended alignment as a two-player game in which a creator evolves the prompt distribution while a solver learns on it, implicitly regularizing prompt shift;

Align-Pro gives a principled objective to optimize a prompter distribution with explicit KL terms (Ye et al., 2024; Trivedi et al., 2025). Complementary strands stabilize optimization or reweight data from the policy side (e.g., StablePrompt, WPO), yet they do not directly constrain the environmental statistics over queries (Kwon et al., 2024; Zhou et al., 2024). In contrast, mainstream PO for RLHF focuses on action regularization via a Policy-KL budget to an SFT reference (Schulman et al., 2015b; 2017; Ouyang et al., 2022), leaving the input/query process comparatively unconstrained. Empirically, even under a fixed Policy-KL budget, the input environment keeps drifting: the batch-estimated Query-KL rises steadily throughout training while the Policy-KL on responses remains nearly flat (Figure 1). This demonstrates that constraining only the action distribution fails to stabilize the input/query process, leaving environment non-stationarity unaddressed.

**In this paper.** We treat queries as part of the environment and make environment statistics a first-class object in the training objective. We introduce **Query-KL regularization (QKL)**, a plug-in penalty on the divergence between the current empirical query sampler and a chosen reference sampler, explicitly limiting inter-round drift of the training environment while leaving the action space free to explore. In parallel, we propose a lightweight query reweighting scheme that reduces estimator variance and improves robustness under high-temperature decoding—where LLMs are especially sensitive to the long tail of decoding distributions (Holtzman et al., 2020; Wang et al., 2023). Both components are model- and optimizer-agnostic and drop into PPO/DPO-style implementations with minimal changes. Figure 2 sketches ERPO: on

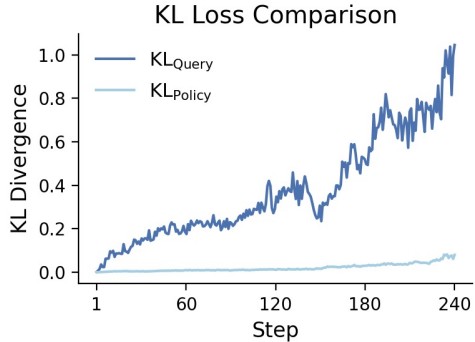

Figure 1: KL losses during GRPO training. The Query-KL (dark) rises while the Policy-KL (light) stays low, showing action-only KL does not stabilize the query process.

top of GRPO we replace the usual Policy-KL with a pre-computed Query-KL, and during advantage computation we weight the within-query samples by the query's occurrence probability, yielding an environment-aware update while preserving action-side exploration.

**Contributions.** We make four main contributions. (1) **Query-environment control:** We treat queries as part of the environment and stabilize training by combining *Query-KL (QKL)* to bound query drift with batch self-normalized query weights to reduce variance and tame high-temperature behavior. (2) **Drop-in practicality:** The method is optimizer-agnostic and adds only a QKL term plus per-batch reweighting on top of GRPO/PPO-style pipelines with minimal changes. (3) **Stability evaluation:** We assess RL stability via *multi-temperature* sampling paired with a *multi-metric* suite (Pass@k, Pass@1, Avg@k), enabling comprehensive capability and robustness evaluation. (4) **Empirical gains:** Across diverse reasoning benchmarks, the approach consistently improves accuracy.

## 2 RELATED WORKS

### 2.1 REINFORCEMENT LEARNING WITH VERIFIABLE REWARDS (RLVR)

Reinforcement Learning with Verifiable Rewards represents a paradigm shift from traditional RLHF approaches by leveraging automatically verifiable outcomes rather than human preference annotations. This approach is particularly powerful for domains where ground truth can be objectively determined, such as mathematical reasoning, code generation, and logical problem solving. Models like AlphaCode (Li et al., 2022) and recent mathematical reasoning (Jeannotte & Kieran, 2017; Xia et al., 2025) systems leverage execution results and correctness verification as direct reward signals, eliminating the need for expensive human annotation.

Process Reward Models (PRMs) have emerged as a sophisticated extension of RLVR, where intermediate steps in reasoning processes are evaluated and rewarded based on their correctness (Uesato et al., 2022; Lightman et al., 2023). Recent developments include tool-augmented reasoning

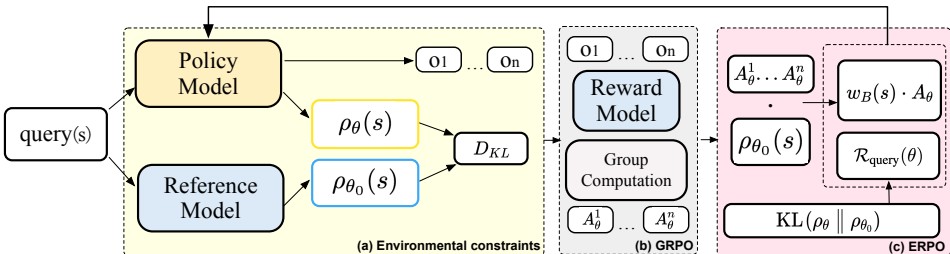

Figure 2: **The Proposed ERPO Overview.** (a) For each query, the policy and reference induce current and reference query samplers, and we pre-compute a Query-KL to penalize environment drift. (b) For each query, the policy samples a response group scored by the reward model to produce the standard GRPO learning signal. (c) On top of GRPO we replace response-KL with pre-computed Query-KL and weight within-query advantages by the query's occurrence probability, yielding an environment-aware update.

systems (Schick et al., 2023) and self-verification approaches (Kojima et al., 2022), which combine language models with external verification tools to enable automatic reward computation for broader task domains.

While RLVR provides scalable and consistent training signals compared to subjective human preferences, it introduces unique challenges in handling high variance from sparse rewards and potential reward hacking behaviors. These stability issues motivate the need for robust training methodologies that can effectively leverage verifiable rewards while maintaining training stability.

### 2.2 REINFORCEMENT LEARNING STABILITY IN LANGUAGE MODEL TRAINING

The stability of reinforcement learning algorithms in language model training has become a critical research area due to unique challenges posed by discrete action spaces, large parameter spaces, and complex reward landscapes (Sutton et al., 1998). Recent works have identified specific stability issues including reward hacking (Gao et al., 2023) and the alignment tax problem (Dai et al., 2025), where policy optimization can degrade downstream performance while improving target metrics. Distribution shift during training has been recognized as a fundamental source of instability in policy gradient methods (Reddy et al., 2020). In language model contexts, this manifests as shifts in the query distribution during training, leading to high variance in gradient estimates and potential policy collapse (Wen et al., 2024). Existing approaches primarily focus on action-space regularization through trust region methods (Schulman et al., 2015a) and KL divergence penalties between current and reference policies.

Despite progress in understanding RL stability issues, there remains a notable gap in explicitly managing the input query distribution during training. Most current approaches focus on output regularization rather than addressing environmental shifts at the input level, leaving query distribution management as an underexplored avenue for improving training stability.

## 3 PRELIMINARIES

### 3.1 NOTATION AND RLVR SETTING

We consider a standard generative–verify (RLVR) setting for training a large language model (LLM) with parameters $\theta$. Given a query (or state) $s \in \mathcal{S}$, the LLM defines a conditional distribution over responses (or actions) $a \in \mathcal{A}$, denoted by $\pi_\theta(a \mid s)$. After generating a response, a verifier computes a scalar reward $g(s, a) \in \mathbb{R}$, which can be a continuous score or a transformed binary signal (e.g., correctness of a solution).

Throughout the paper we use the following notation:

- $s \in \mathcal{S}$: query / state (e.g., a math problem or user prompt);
- $a \in \mathcal{A}$: response / action generated by the LLM;

- $\pi_\theta(a \mid s)$: policy implemented by the LLM;
- $g(s, a)$: reward returned by the verifier;
- $\rho_\theta(s)$: *policy-induced query distribution*, defined below;
- $\rho_{\theta_0}(s)$: reference query distribution, e.g., induced by an initial SFT policy $\pi_{\theta_0}$.

The underlying RL objective can be written as the expected reward

$$J_{\text{RL}}(\theta) = \mathbb{E}_{s \sim \rho_\theta, \, a \sim \pi_\theta(\cdot \mid s)} \big[ g(s, a) \big]. \tag{1}$$

In practice, group-based or advantage-based variants of equation 1 are often used (e.g., GRPO(Shao et al., 2024), RLOO(Ahmadian et al., 2024), DAPO(Yu et al., 2025)); our method is compatible with any such policy-gradient estimator and does not depend on a particular choice. In the experiments we instantiate the policy-gradient surrogate with *GRPO* (Shao et al., 2024), which defines a group-relative advantage from $K$ sampled responses for each query $s$:

$$A_\theta^{\text{GRPO}}\big(s, a^{(k)}\big) = \frac{g\big(s, a^{(k)}\big) - \text{mean}(\mathbf{g})}{\text{std}(\mathbf{g})}. \tag{2}$$

This $A_\theta^{\text{GRPO}}$ is a concrete instance of the generic advantage $A_\theta(s, a)$ in the policy-gradient identity and will be plugged into our ERPO objective in Section 4.

### 3.2 POLICY-INDUCED QUERY DISTRIBUTION AND ENVIRONMENT DRIFT

Formally, we view the training process as interacting with an environment whose states are queries $s$. Updating the policy $\pi_\theta$ not only changes how the model responds to a given query $s$, but also changes how likely different queries are to appear in the training batches. This leads to *policy-induced environment drift*: as $\theta$ evolves, the effective environment seen by the learner shifts from its initial state.

To quantify this drift, we compare $\rho_\theta$ with a fixed reference distribution $\rho_{\theta_0}$, typically taken to be the query distribution induced by the initial SFT model $\pi_{\theta_0}$. We define the environment shift as the forward KL divergence

$$\text{EnvShift}(\theta) := \text{KL}(\rho_\theta \,\|\, \rho_{\theta_0}) = \mathbb{E}_{s \sim \rho_\theta} \left[ \log \frac{\rho_\theta(s)}{\rho_{\theta_0}(s)} \right]. \tag{3}$$

### 3.3 QUERY LIKELIHOOD IN AUTOREGRESSIVE LLMS

Our method relies on the likelihood of a query $s$ under the current model. For an autoregressive LLM, any token sequence $s = (s_1, \ldots, s_T)$ is assigned a probability

$$P_\theta(s) = \prod_{t=1}^{T} P_\theta(s_t \mid s_{<t}), \qquad \log P_\theta(s) = \sum_{t=1}^{T} \log P_\theta(s_t \mid s_{<t}), \tag{4}$$

where $s_{<t}$ denotes the prefix $(s_1, \ldots, s_{t-1})$. Importantly, this likelihood is well-defined for *any* query sequence $s$, regardless of whether $s$ is sampled on-policy from $\rho_\theta$, drawn from a static dataset, or written by humans. Computing $\log P_\theta(s)$ only requires a forward pass of the LLM and does not depend on how $s$ is obtained.

## 4 METHOD

We now instantiate our Environment-Regularized Policy Optimization (ERPO) framework. Building on the preliminaries in Section 3, we (i) define the population-level ERPO objective and its empirical approximation, (ii) specify the query-level reweighting and query-level KL regularizer that stabilize the environment, and (iii) show how ERPO can be combined with any policy-gradient (PG) algorithm via a unified surrogate loss. The convergence of stochastic gradient descent (SGD) on the population objective is analyzed in Appendix F.

### 4.1 POPULATION OBJECTIVE AND EMPIRICAL LOSS

Recall from Section 3 that the underlying RL objective is the expected return equation 1, and that the per-query expected return is

$$\bar{g}_\theta(s) = \mathbb{E}_{a \sim \pi_\theta(\cdot|s)}\big[g(s, a)\big], \tag{5}$$

as in equation 1–equation 5. To control policy-induced environment drift, we introduce a query-level regularizer $\mathcal{R}_{\text{query}}(\theta)$ that penalizes the forward KL between the policy-induced query distribution $\rho_\theta$ and a reference distribution $\rho_{\theta_0}$ (cf. Section 3.2).

Our environment-regularized population objective is

$$J_{\text{ERPO}}(\theta) := \mathbb{E}_{s \sim \rho_\theta}\big[\bar{g}_\theta(s)\big] - \alpha \mathcal{R}_{\text{query}}(\theta), \tag{6}$$

where $\alpha > 0$ controls the strength of environment regularization. The convergence analysis in Appendix F is stated in terms of this population-level objective: under standard smoothness and variance assumptions, SGD on $J_{\text{ERPO}}(\theta)$ converges to a stationary point and yields an explicit bound on $\mathcal{R}_{\text{query}}(\theta)$ (hence on environment drift).

In practice we only observe a mini-batch $B = \{s_i\}_{i=1}^m$ of queries sampled from $\rho_\theta$, and thus approximate the outer expectation in equation 6 via a Monte Carlo estimator. Since queries play the role of environment states in our formulation, we further introduce query-level importance weights $w_B(s)$ to reweight the empirical query distribution so as to better capture environment variation (see Section 4.2 for details). Using these weights, we obtain the empirical objective

$$\widehat{J}_B(\theta) := \frac{1}{m} \sum_{s \in B} w_B(s)\, \bar{g}_\theta(s) - \alpha \widehat{\mathcal{R}}_{\text{query}}(\theta), \tag{7}$$

where $\widehat{\mathcal{R}}_{\text{query}}(\theta)$ is a batch estimator of the query-level KL (Section 4.2). Our training loss is the negative of this objective, realized via a PG surrogate described in Section 4.3.

### 4.2 QUERY REWEIGHTING AND QUERY-LEVEL KL

This subsection specifies (i) how we choose the batch weights $w_B(s)$ and (ii) how we instantiate the query-level KL regularizer. Both constructions directly connect to the preliminaries on query likelihood and environment drift in Sections 3.2–3.3.

**Query-weighted objective.** Recall that the policy-induced query distribution $\rho_\theta$ drifts during training (Section 3.2). In Eq. equation 7, the expectation is taken with respect to this query distribution, and in our setting we explicitly treat queries as the source of environment variation. Accordingly, beyond the KL regularizer, we further stabilize training by assigning query-level importance weights that focus gradient updates on *high-confidence* queries while down-weighting outliers that may induce large gradient variance.

Using the sequence log-likelihood $\ell_\theta(s) = \log P_\theta(s)$ from equation 4, we define the batch weight

$$w_B(s) = \frac{\bar{\ell}_B}{\ell_\theta(s)} > 0, \qquad \bar{\ell}_B = \frac{1}{m} \sum_{s' \in B} \ell_\theta(s')\ (< 0), \tag{8}$$

where higher-likelihood queries receive larger weights. This construction preserves the likelihood-based ordering while compressing the dynamic range compared to exponential importance weights, thereby reducing variance. The weights are computed with **stop-gradient** during backpropagation. A detailed derivation from a self-normalized substitute distribution is given in Appendix G.

Substituting the batch weights into equation 6 yields the *query-weighted* Monte-Carlo objective

$$\widehat{J}_B^{\text{reweight}}(\theta) := \frac{1}{m} \sum_{s \in B} w_B(s)\, \bar{g}_\theta(s), \tag{9}$$

which corresponds to the first term in equation 7.

**Query-level KL regularization.** As discussed in Section 3.2, the policy-induced query distribution $\rho_\theta$ co-evolves with the policy $\pi_\theta$, leading to environment drift. We regularize this drift using a query-level forward KL to a fixed or slowly updated reference $\theta_0$:

$$\mathcal{R}_{\text{query}}(\theta) := \text{KL}\big(\rho_\theta \,\big\|\, \rho_{\theta_0}\big) = \mathbb{E}_{s\sim\rho_\theta}\big[\log\rho_\theta(s) - \log\rho_{\theta_0}(s)\big]. \tag{10}$$

This penalizes forgetting probability mass under $\rho_{\theta_0}$ while leaving the action-level policy unconstrained, thereby preserving response-side exploration.

In practice we do not have closed-form access to $\rho_\theta(s)$, but we can estimate the gradient of equation 10 via the "K3" approximation to the KL divergence gradient (Schulman, 2020). Concretely, on a batch $B$ we approximate

$$\widehat{\mathcal{R}}_{\text{query}}(\theta) \approx \frac{1}{m}\sum_{s\in B}\big[\log P_\theta(s) - \log P_{\theta_0}(s)\big], \tag{11}$$

using the query likelihoods $P_\theta(s)$ and $P_{\theta_0}(s)$ from equation 4, and treat $\log P_{\theta_0}(s)$ as constant during backpropagation. The resulting estimator is a standard Monte-Carlo approximation to the K3 surrogate gradient of the forward KL. This is the second term in our empirical objective equation 7.

### 4.3 PG-COMPATIBLE SURROGATE AND ERPO INSTANTIATION

We now describe how to realize stochastic gradients of the ERPO objective using a generic PG-family surrogate. For any policy-gradient algorithm, we can approximate the gradient of the per-query return in equation 5 with a surrogate of the form

$$\nabla_\theta \bar{g}_\theta(s) \approx \mathbb{E}_{a\sim\pi_\theta(\cdot\mid s)}\big[u_\theta(s,a)\,A_\theta^\star(s,a)\nabla_\theta\log\pi_\theta(a\mid s)\big], \tag{12}$$

where $A_\theta^\star(s,a)$ is an algorithm-specific advantage and $u_\theta(s,a)$ is an action-level weight. For example, $u_\theta \equiv 1$ and $A_\theta^\star$ equal to the group-relative advantage from equation 2 recover GRPO; clipped probability ratios recover PPO; and $A_\theta^\star$ equal to the sample-wise reward recovers REINFORCE.

Combining the query-weighted objective equation 9 with the query-level KL regularizer equation 10, and replacing $\bar{g}_\theta(s)$ by the PG surrogate equation 12, we obtain the following mini-batch loss:

$$\mathcal{L}_{\text{PG-family}}(\theta) := -\frac{1}{m}\sum_{s\in B}w_B(s)\frac{1}{K}\sum_{a\in\mathcal{G}(s)}u_\theta(s,a)\,A_\theta^\star(s,a) \,+\, \alpha\,\widehat{\mathcal{R}}_{\text{query}}(\theta), \tag{13}$$

where $\mathcal{G}(s)$ is the set of $K$ responses sampled for query $s$. During backpropagation, the outer weights $w_B(s)$ are treated as constants (stop-gradient), so $\nabla_\theta\mathcal{L}_{\text{PG-family}}(\theta)$ is an unbiased stochastic gradient of the ERPO objective equation 6 under standard assumptions; see Appendix F.

**Instantiation with GRPO.** For experiments we instantiate equation 13 with a GRPO-style group-relative baseline. For each query $s$ we sample $\mathcal{G}(s) = \{a^{(k)}\}_{k=1}^K$, compute the group-relative advantage $A_\theta^{\text{GRPO}}(s,a)$ as in equation 2, take $u_\theta \equiv 1$, and optimize

$$\mathcal{L}_{\text{ERPO}}(\theta) := -\frac{1}{m}\sum_{s\in B}w_B(s)\frac{1}{K}\sum_{a\in\mathcal{G}(s)}A_\theta^{\text{GRPO}}(s,a)\log\pi_\theta(a\mid s) \,+\, \alpha\,\widehat{\mathcal{R}}_{\text{query}}(\theta). \tag{14}$$

All other engineering details from GRPO (reward normalization, group size $K$, sampling temperature, etc.) remain unchanged. Our contribution is orthogonal: we *replace* the per-query outer weight by $w_B(s)$ and *add* the query-level KL term, yielding a simple drop-in modification that can be applied to other PG algorithms as well.

## 5 EXPERIMENTS

### 5.1 EXPERIMENTAL SETUP

**Training** We conduct experiments on mathematical reasoning tasks using Level 3–5 problems from the MATH dataset (Hendrycks et al., 2021), totaling approximately 8.5K examples. These are used to evaluate our proposed ERPO method, in comparison with the vanilla GRPO baseline. As described in Appendix A, the model must wrap its intermediate reasoning in `<think></think>` tags, and place the final answer inside `\boxed {}`.

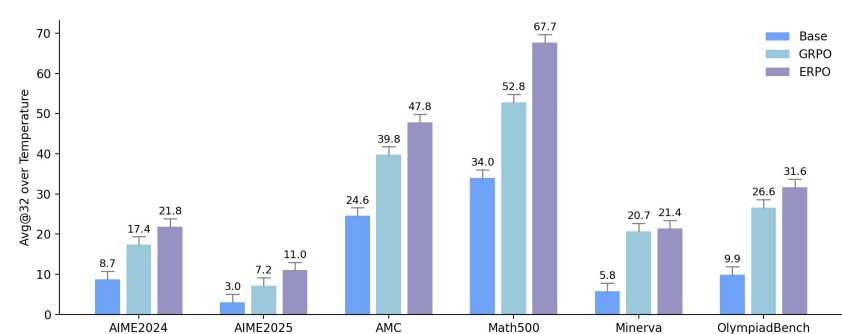

Figure 3: Avg@32 over Sampling Temperatures on Mathematical Reasoning Tasks

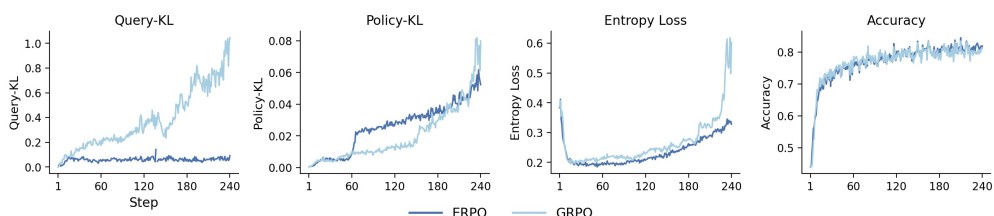

Figure 4: Training Dynamics on ERPO

**Evaluation** We follow standard practice and assess performance on six widely used benchmarks: AIME24, AIME25, AMC, MATH500 (Hendrycks et al., 2021), Minerva (Lewkowycz et al., 2022), and OlympiadBench (He et al., 2024). Prior work typically reports Avg@K (Yu et al., 2025), Pass@1 (Liu et al., 2025), and Pass@K (Hao et al., 2025) after RLVR training, often without specifying or controlling the inference-time sampling temperature. This omission can substantially affect reported performance and render results across studies not directly comparable. In preliminary experiments, we found that inference-time sampling temperature has a significant impact on performance, and that the effect intensifies as training progresses. To control for this factor, we fix the number of training steps across all models and evaluate at temperatures from 0.1 to 1.5; performance is then aggregated over this range.

**Implementation Details** We conduct all experiments using the EasyR1 framework (Zheng et al., 2025), training the Qwen2.5-Math-7B and Qwen2.5-32B model (Yang et al., 2024; Qwen et al., 2025) with both GRPO and ERPO algorithms. Following prior work (Liu et al., 2025), we set the maximum sequence length to 3K tokens. For each problem, we sample eight responses at an inference temperature of 1.0. The rollout batch size is set to 512, and the update batch size to 128, for a total of 240 training steps. Token-level loss is applied throughout training. To ensure a fair comparison, we adopt the default KL divergence coefficient of 0.01.

## 5.2 MAIN RESULTS

Figure 3 summarizes Avg@32 accuracy on six mathematical reasoning benchmarks, averaged over sampling temperatures from 0.1 to 1.5. ERPO consistently outperforms GRPO, with gains of up to 14.9% and an overall average improvement of 6.2%, highlighting its enhanced capability. Table 1 presents the detailed results for each benchmark, grouped by evaluation metric (e.g., Pass@1, Pass@K).

For both GRPO and ERPO, the prompts are identical to those used during training, whereas the Qwen base model adopts the default configuration from Dr.GRPO (Liu et al., 2025) to ensure optimal performance. Consistent with the aggregated results in Figure 3 and Table 1, ERPO surpasses GRPO across all evaluation metrics, achieving improvements of 6.2% in Avg@32, 3.64% in Pass@32, and 5.69% in Pass@1. We also applied the concept of ERPO to other RLVR algorithms and observed similarly effective gains; details are provided in Appendix E.

Table 1: Performance comparison across mathematical reasoning benchmarks. Best results per column are highlighted in bold.

| Method | AIME24 | AIME25 | AMC | MATH500 | Minerva | Olympiad | Avg. |
|---|---|---|---|---|---|---|---|
| **Mean Avg@32** | | | | | | | |
| Base | 0.087 | 0.030 | 0.246 | 0.340 | 0.058 | 0.099 | 0.143 |
| GRPO | 0.174 | 0.072 | 0.398 | 0.528 | 0.207 | 0.266 | 0.274 |
| ERPO | **0.218** | **0.110** | **0.478** | **0.677** | **0.214** | **0.316** | **0.336** |
| **Mean Pass@32** | | | | | | | |
| Base | 0.373 | 0.206 | 0.674 | 0.764 | 0.349 | 0.411 | 0.463 |
| GRPO | 0.471 | 0.287 | 0.768 | 0.850 | **0.516** | 0.558 | 0.575 |
| ERPO | **0.509** | **0.342** | **0.820** | **0.904** | 0.500 | **0.593** | **0.611** |
| **Mean Pass@1** | | | | | | | |
| Base | 0.090 | 0.038 | 0.264 | 0.342 | 0.062 | 0.099 | 0.149 |
| GRPO | 0.169 | 0.084 | 0.398 | 0.533 | 0.201 | 0.263 | 0.275 |
| ERPO | **0.207** | **0.091** | **0.477** | **0.679** | **0.217** | **0.320** | **0.332** |

## 5.3 TRAINING DYNAMICS

Figure 4 illustrates the training dynamics of the ERPO method. For both approaches, the sampling accuracy on the training set remains largely consistent; however, their divergence from the reference model exhibits markedly different trajectories.

In GRPO, constraints are imposed on the action distribution, causing the query distribution to drift away from the reference model at a substantially faster rate. Consequently, the KL divergence at the query level is an order of magnitude greater than at the policy level.

This imbalance leads to pronounced discrepancies in performance between the training and evaluation datasets. In contrast, ERPO applies constraints directly to the query distribution and adjusts the loss according to the probability of the given problem. This design both limits the degree of divergence from the reference model during training and, by leveraging the independence between the problem and the response, allows unconstrained exploration at the policy level. A quantitative analysis can be found in Appendix C. As a result, ERPO achieves superior generalization performance on general problems.

## 5.4 ANALYSIS

**Ablation Study**   We conduct ablation studies on the MATH500 benchmarks to assess reasoning efficiency. Table 2 summarizes the results for several commonly used sampling temperatures. Figure 5 further provides the complete performance–temperature variation curves across different experimental settings, along with the corresponding training dynamics.

**Mechanisms**   Without modifying other hyperparameters, replacing the policy-based KL divergence with query-based KL divergence yields the best overall performance[1], with an average improvement of 15.9% over GRPO. In contrast, GRPO with policy-based KL divergence shows its highest performance only at a temperature of 1.0 (see Figure 5(a)).

To further investigate, an ablation study is conducted on the two mechanisms of ERPO with their effects evaluated using KL divergence and entropy (Table 3 and Figure 5(d)). The term $w_{B(s)}$ downweights gradients from low-probability queries, which often lead to low-probability responses, thereby increasing gradient variance and entropy (Quantitative analysis is provided in Appendix D). Introducing $w_{B(s)}$ allows sufficient training while concurrently reducing the policy KL divergence.

---

[1]We also experimented with completely removing all KL divergence constraints, which resulted in the training process failing to converge.

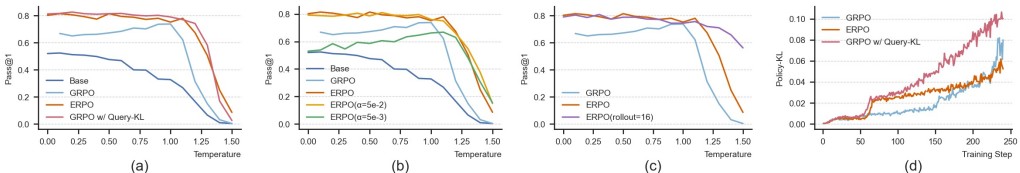

Figure 5: Pass@1 accuracy and training dynamics under different settings: (a)–(c) Model performance at various temperatures on MATH500; (d) Policy-KL divergence variation with GRPO using only Query-KL.

Different regularization strengths $\alpha$ also exert a significant influence on performance. As the constraint strength increases (e.g., $\alpha = 5 \times 10^{-2}$), the model achieves further improvements in overall performance (see Table 2). It is worth noting that we did not conduct an exhaustive search for the optimal $\alpha$; instead, we retained the default value to ensure a relatively fair comparison.

Table 2: Performance Comparison Under Different Experimental Settings

| Base Model | Method | $\alpha$ | w(s) | Rollout Count | Temperature Metrics | | | | | |
|---|---|---|---|---|---|---|---|---|---|---|
| | | | | | 0.1 | 0.6 | 1 | 1.5 | $\leq$1.0 | 1.2-1.5 |
| **Baseline** | — | — | — | — | 52.40 | 46.80 | 32.80 | 0.40 | 44.44 | 6.15 |
| **Qwen-7B** | GRPO | $1 \times 10^{-2}$ | — | 8 | 66.80 | 68.40 | 73.80 | 0.40 | 68.80 | 12.50 |
| | | $1 \times 10^{-2}$ | — | 16 | 73.00 | 79.20 | 75.00 | 10.60 | 75.22 | 39.75 |
| | ERPO | $1 \times 10^{-2}$ | ✓ | 8 | 79.40 | 80.60 | 75.20 | 8.60 | 78.74 | 37.90 |
| | | $1 \times 10^{-2}$ | — | 8 | **81.60** | **81.60** | **79.00** | 2.60 | **80.90** | 38.00 |
| | | $5 \times 10^{-3}$ | ✓ | 8 | 53.80 | 60.60 | 66.20 | 15.40 | 59.94 | 39.30 |
| | | $5 \times 10^{-2}$ | ✓ | 8 | 78.80 | 81.00 | 76.00 | 15.00 | 79.00 | 43.35 |
| | | $1 \times 10^{-2}$ | ✓ | 16 | 80.40 | 78.80 | 74.40 | **56.20** | 77.82 | **66.25** |
| | GRPO* | $1 \times 10^{-2}$ | ✓ | 8 | 78.20 | 76.80 | 71.20 | 7.60 | 76.14 | 23.79 |
| **Qwen-32B** | GRPO | $1 \times 10^{-2}$ | — | 8 | 81.60 | 82.40 | 81.20 | 25.20 | 81.62 | 57.20 |
| | ERPO | $1 \times 10^{-2}$ | ✓ | 8 | **85.00** | **84.80** | **83.60** | **80.80** | **84.60** | **82.80** |

**Note:** Best results per column are highlighted in **bold** (separate for 7B and 32B). The columns $\leq$**1.0** and **1.2–1.5** show the mean accuracy (Acc) over the corresponding temperature ranges. The $w(s)$ column indicates whether query-rweighting is applied (✓) or not (—). GRPO* denotes the GRPO algorithm using only query-reweighting.

Table 3: Influence of Query-KL and Query-Reweighting on Training Stability

| Method | Query-KL | Policy-KL | Entropy |
|---|---|---|---|
| GRPO (Avg@3) | 0.9679 | 0.0601 | 0.5063 |
| GRPO w/$w_{B(s)}$ | 0.5933 | **0.0113** | **0.2782** |
| GRPO w/Query-KL | **0.0041** | 0.1001 | 0.5674 |
| ERPO (Avg@3) | 0.0828 | 0.0728 | 0.4244 |

**Rollouts** We also analyze the effect of the number of samples per query. By increasing the sampling number to 16, we achieve the best performance, with the average Pass@1 rising to 74.6%. A higher sampling count also significantly improves sampling stability at high temperatures (see Table 2), without a noticeable increase in divergence from the reference model. Moreover, increasing the sampling count facilitates ERPO-based models in acquiring the correct reasoning format more effectively. [2]

**Long-term Training** To assess the stability of long-term RL training, we scale the training steps up to 1K and monitor changes in model performance over time. As shown in the fig-

---

[2]Across multiple experiments, the GRPO method consistently failed to capture the desired output format. Consequently, for all experiments, we report only the answer accuracy.

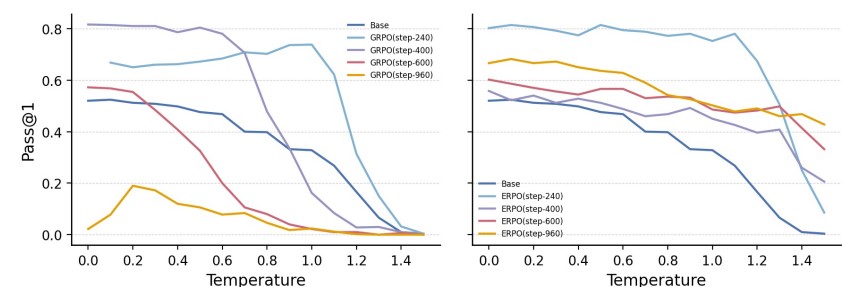

Figure 6: Performance Variation Across Training Steps

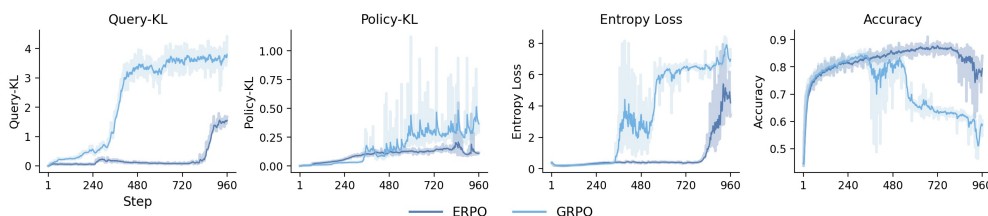

Figure 7: Training Dynamics on Long-term RL

ure 6 and 7, GRPO remains stable for sampling temperatures below 1.0 until approximately 240 steps (epoch=15). However, a pronounced performance degradation is first observed in the high-temperature sampling regime after 400 steps, and subsequently propagates to encompass sampling across all temperatures as the steps increase.

In contrast, ERPO exhibits a modest performance decline; however, the overall deterioration is substantially smaller, and its performance even improves within the high-temperature range. Figure 6 presents the complete training trajectories for both GRPO and ERPO. Although ERPO is not entirely immune to the collapse phenomenon that may occur during extended training—manifested as a sudden increase in entropy and a loss of sampling capability—it consistently outperforms vanilla GRPO and achieves a comparable degree of policy distribution constraint without relying on an explicit policy-based KL divergence term.

## 6 CONCLUSION

By analyzing the coupling between the environment and the policy space in large language models, we decouple parameter regularization from the optimization objective during training. Specifically, we employ query-level KL divergence to indirectly constrain the distance between the policy model and the reference model. To prevent the model from prematurely converging to suboptimal solutions, we weight the advantage by the occurrence probability of each query. Experiments across multiple mathematical reasoning benchmarks demonstrate that the proposed ERPO method can achieve comparable KL divergence control without explicit policy regularization, while delivering superior performance. Furthermore, by sampling at different temperatures, we examine the evolution of sampling capability over long-term RL training, providing additional evidence of ERPO's stability during training.

## REPRODUCIBILITY STATEMENT

We use open-source datasets for both training and testing, and conduct all experiments on an NVIDIA A100 GPU cluster. The complete environment configuration and step-by-step instructions for reproducing our results are openly available at: `https://anonymous.4open.science/r/ERPO-5B0C/`

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

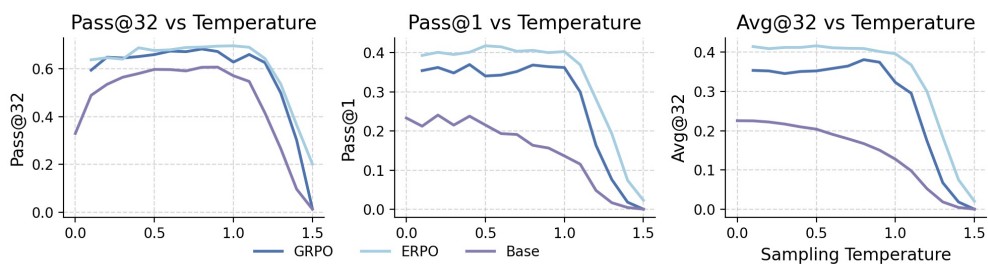

Figure 8: Variation of Metrics with Temperature

## A  PROMPT

*{{ content | trim }} You FIRST think about the reasoning process*
*as an internal monologue and then provide the final answer.  The*
*reasoning process MUST BE enclosed within <think> </think> tags.*
*The final answer MUST BE put in \boxed {}.*

## B  VARIATION OF METRICS WITH TEMPERATURE

Figure 8 illustrates the model performance across different evaluation metrics and sampling temperatures. Our approach reduces the performance gap between different sampling temperatures, while increasing the likelihood of sampling correct outputs.

## C  ANALYSIS OF REWARD HACKING

In the course of our experiments, we observed a severe reward hacking phenomenon when training with the baseline GRPO method. Specifically, while the model consistently achieved high rewards on the training data, its performance on the evaluation set often plateaued or even degraded during the later stages of optimization. This pronounced discrepancy suggests that the model overfits to the specific characteristics of the reward signal during training sampling, failing to generalize to the standard decoding distribution used during inference.

To quantitatively investigate this issue, we monitored the Train–Evaluation Consistency throughout the training process. We periodically evaluated both the training accuracy and the evaluation accuracy every ten optimization steps. To ensure the robustness of our inference metrics, evaluation was conducted using vLLM under two different Tensor Parallelism settings (TP1 and TP2), a factor which has been shown in previous work (Yao et al., 2025) to impact model performance.

Table 4: **Quantification of Reward Hacking via Train–Inference Gap.** The table compares the average accuracy during training sampling versus inference decoding. A larger gap indicates severe reward hacking (overfitting to training dynamics). ERPO reduces this gap by $\approx 51\%$, demonstrating robust generalization.

| Method | Avg. Train Acc | Avg. Eval@TP1 | Avg. Eval@TP2 | Gap@TP1 | Gap@TP2 | Avg. Gap |
|---|---|---|---|---|---|---|
| GRPO | 77.3 | 69.8 | 70.1 | 7.5 | 5.5 | 6.47 |
| ERPO | 77.1 | 74.1 | 73.8 | 3.0 | 3.3 | 3.14 |
| **Improvement** | – | – | – | ↓ 4.5 (60%) | ↓ 2.2 (40%) | ↓ 3.33 (51%) |

Table 4 summarizes the average performance gap across six key checkpoints (Steps 40, 80, 120, 160, 200, and 240). The results confirm our hypothesis:

- GRPO exhibits a substantial average gap of 6.47%, indicating a significant misalignment between training and inference behaviors. Notably, as shown in the detailed trajectories in Table 5, GRPO's evaluation accuracy drops sharply at Step-240 (from $\approx 75\%$ to 58.4%) despite maintaining high training accuracy, a classic signature of reward hacking.

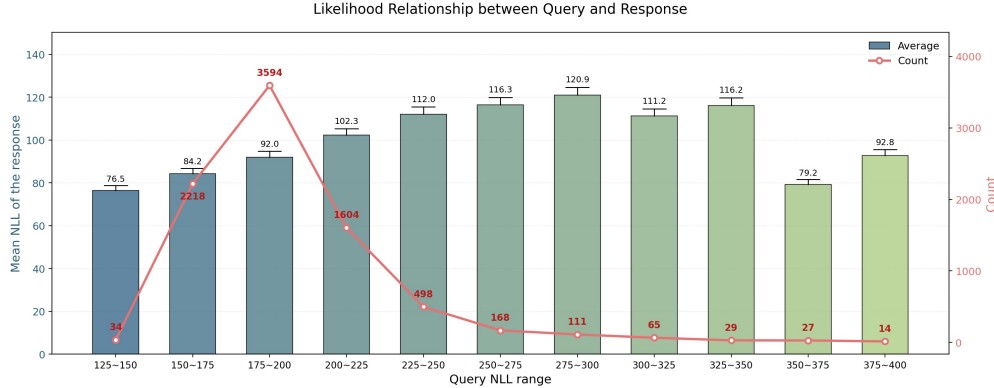

Figure 9: Likelihood Relationship between Query and Response

- ERPO, in contrast, demonstrates superior consistency. It reduces the average Train–Eval gap by approximately 51% (from 6.47% to 3.14%).

Table 5: **Trajectory of Train vs. Inference Accuracy.** Detailed performance recorded at 40-step intervals. Note the divergence in GRPO at Step-240, where Eval accuracy drops significantly while Train accuracy remains high—a clear sign of reward hacking. ERPO maintains consistency throughout.

| Model / Metric | Step-0 | Step-40 | Step-80 | Step-120 | Step-160 | Step-200 | Step-240 | Avg. |
|---|---|---|---|---|---|---|---|---|
| *GRPO (Baseline)* | | | | | | | | |
| Train Acc | 44.48 | 76.41 | 76.09 | 78.29 | 78.95 | 79.44 | 76.70 | 72.91 |
| Eval@TP1 | 31.20 | 73.20 | 76.00 | 73.40 | 72.20 | 73.60 | 58.40 | 65.43 |
| Eval@TP2 | 30.60 | 74.00 | 74.80 | 74.20 | 76.60 | 75.80 | 66.20 | 67.46 |
| *ERPO (Ours)* | | | | | | | | |
| Train Acc | 44.55 | 75.63 | 76.53 | 78.66 | 80.63 | 78.41 | 81.46 | 73.70 |
| Eval@TP1 | 31.20 | 74.00 | 77.20 | 77.00 | 78.40 | 78.60 | 78.40 | 70.69 |
| Eval@TP2 | 30.60 | 72.60 | 77.00 | 76.60 | 77.40 | 78.60 | 80.20 | 70.43 |

This significant reduction in the performance gap indicates that ERPO effectively regularizes the training process, preventing the model from exploiting spurious patterns in the reward function and ensuring that improvements in training translate reliably to inference performance.

## D CORRELATION ANALYSIS BETWEEN QUERY AND RESPONSE PROBABILITIES

We sampled over 8K questions from the training dataset at a temperature of 1.0 and independently computed the negative log-likelihood (NLL) for both the prompt and response parts:

$$\text{NLL}(x) = -\sum \log p(x)$$

where $x$ denotes the generation probabilities of tokens in the prompt or response. The resulting histogram is shown in Figure 9. We observe a positive correlation between the NLL of the prompt and that of the response. For 95% of the training samples (where the NLL of the prompt is less than 300), the correlation coefficient is close to 1. Low-probability responses appearing in positive samples contribute to an increase in entropy during training.

## E ERPO ON DIFFERENT ALGORITHMS

Additional experiments were conducted on DAPO(Yu et al., 2025) and RLOOAhmadian et al. (2024) with and without the global KL divergence constraint, yielding absolute improvements of 10.24% and 2.28% at temperatures below 1.0, respectively (see Table 6). These findings demonstrate that the proposed method can achieve significant gains when applied to other RLVR algorithms.

Table 6: Performance Comparison Under Different Experimental Settings

| Base Model | Method | D_KL | Rollout | Temperature | | | | | |
|---|---|---|---|---|---|---|---|---|---|
| | | | | 0.1 | 0.6 | 1 | 1.5 | ≤1.0 | 1.2-1.5 |
| Baseline | – | – | – | 52.4 | 46.8 | 32.8 | 0.4 | 44.44 | 6.15 |
| Qwen-7B | GRPO | Policy | 8 | 66.8 | 68.4 | 73.8 | 0.4 | 68.8 | 12.5 |
| | | | 16 | 73.0 | 79.2 | 75.0 | 10.6 | 75.22 | 39.75 |
| | ERPO | Query | 8 | 79.4 | 80.6 | 75.2 | 8.6 | 78.74 (+9.94) | 37.9 |
| | | | 16 | 80.4 | 78.8 | 74.4 | 56.2 | 77.82 (+2.6) | 66.25 |
| | DAPO | – | 8 | 62.0 | 77.4 | 65.4 | 5.6 | 68.16 | 14.0 |
| | DAPO+ERPO | Query | | 80.2 | 79.4 | 75.8 | 20.2 | 78.4 (+10.24) | 36.93 |
| | RLOO | Policy | 8 | 77.6 | 78.4 | 75.4 | 12.4 | 77.28 | 35.93 |
| | RLOO+ERPO | Query | | 81.2 | 80.4 | 79.4 | 17.6 | 79.56 (+2.28) | 40.8 |
| Qwen-32B | GRPO | Policy | 8 | 81.6 | 82.4 | 81.2 | 25.2 | 81.62 | 57.2 |
| | ERPO | Query | | **85.0** | **84.8** | **83.6** | **80.8** | **84.6** (+2.98) | **82.8** |

**Note:** The columns ≤**1.0** and **1.2–1.5** show the mean accuracy (Acc) over the corresponding temperature ranges. Values in parentheses indicate improvements over baseline methods. Best results per column are highlighted in **bold**, second-best results are underlined.

# F    CONVERGENCE ANALYSIS OF ERPO

In this appendix, we provide a simple theoretical view of Environment-Regularized Policy Optimization (ERPO) as stochastic gradient descent (SGD) on a regularized objective, and give a basic convergence guarantee to a stationary point under standard assumptions. We also show that the Query-KL term implicitly bounds environment drift measured in KL divergence.

Throughout this appendix, let $s$ denote a query (input), and let $\rho_\theta(s)$ denote the query distribution induced by the current policy parameterized by $\theta$. The initial query distribution at the beginning of training is denoted by $\rho_{\theta_0}(s)$. We write the query-weighted gain (e.g., a query-weighted advantage) as $\bar{g}_\theta(s)$. The Query-KL regularizer is defined as

$$R_{\text{query}}(\theta) \triangleq \text{KL}\big(\rho_\theta \,\|\, \rho_{\theta_0}\big), \tag{15}$$

and the corresponding ERPO objective is

$$J_{\text{ERPO}}(\theta) \;=\; \mathbb{E}_{s\sim\rho_\theta}\big[\bar{g}_\theta(s)\big] \;-\; \lambda\, R_{\text{query}}(\theta), \tag{16}$$

with regularization coefficient $\lambda > 0$. We consider the associated loss

$$L(\theta) \;=\; -\mathbb{E}_{s\sim\rho_\theta}\big[\bar{g}_\theta(s)\big] \;+\; \lambda\, R_{\text{query}}(\theta), \tag{17}$$

which ERPO seeks to minimize.

### ERPO AS A REGULARIZED OBJECTIVE

**Proposition 1** (ERPO as a regularized objective)**.** *Consider the ERPO objective*

$$J_{ERPO}(\theta) \;=\; \mathbb{E}_{s\sim\rho_\theta}\big[\bar{g}_\theta(s)\big] \;-\; \lambda\, R_{query}(\theta), \qquad R_{query}(\theta) \triangleq \text{KL}\big(\rho_\theta \,\|\, \rho_{\theta_0}\big), \tag{18}$$

*and its corresponding loss*

$$L(\theta) \;=\; -\mathbb{E}_{s\sim\rho_\theta}\big[\bar{g}_\theta(s)\big] \;+\; \lambda\, R_{query}(\theta). \tag{19}$$

*In the idealized infinite-sample limit, the ERPO update is equivalent to stochastic gradient descent on the regularized loss $L(\theta)$, i.e.,*

$$\theta_{t+1} \;=\; \theta_t \;-\; \eta_t \,\nabla L(\theta_t), \tag{20}$$

*where $\eta_t$ is the learning rate.*

*Proof.* By definition, the loss $L(\theta)$ can be written as

$$L(\theta) = \mathbb{E}_{s \sim \rho_\theta} \Big[ - \bar{g}_\theta(s) \, + \, \lambda \, \ell_{\text{QKL}}(\theta; s) \Big], \tag{21}$$

where $\ell_{\text{QKL}}(\theta; s)$ denotes the per-query contribution of the Query-KL term (for instance, the Monte Carlo integrand of $\text{KL}(\rho_\theta \,\|\, \rho_{\theta_0})$). Let $B$ denote a mini-batch of queries sampled from the training procedure, and consider the empirical loss

$$\widehat{L}_B(\theta) = \frac{1}{|B|} \sum_{s \in B} \Big( - \bar{g}_\theta(s) \, + \, \lambda \, \ell_{\text{QKL}}(\theta; s) \Big). \tag{22}$$

Under standard Monte Carlo sampling assumptions (or in the infinite-sample limit), $\widehat{L}_B(\theta)$ is an unbiased estimator of $L(\theta)$, and its gradient $\nabla \widehat{L}_B(\theta)$ is an unbiased estimator of $\nabla L(\theta)$. Therefore, the ERPO parameter update based on $\nabla \widehat{L}_B(\theta)$ corresponds to stochastic gradient descent on the regularized loss $L(\theta)$. $\qquad\square$

### CONVERGENCE TO A STATIONARY POINT

We now give a simple convergence guarantee under standard nonconvex SGD assumptions.

**Theorem 1** (Convergence to a stationary point of the ERPO loss)**.** *Let $L(\theta)$ be the ERPO loss*

$$L(\theta) = -\mathbb{E}_{s \sim \rho_\theta} \big[ \bar{g}_\theta(s) \big] + \lambda \, R_{query}(\theta), \qquad R_{query}(\theta) = \text{KL}\big( \rho_\theta \,\|\, \rho_{\theta_0} \big). \tag{23}$$

*Assume that:*

1. ***Smoothness.*** *$L(\theta)$ is L-smooth, i.e., its gradient is Lipschitz continuous:*

$$\| \nabla L(\theta) - \nabla L(\theta') \| \, \leq \, L \, \| \theta - \theta' \| \quad \text{for all } \theta, \theta'. \tag{24}$$

2. ***Unbiased stochastic gradients with bounded variance.*** *The mini-batch gradient $g_t$ used by ERPO at iteration $t$ satisfies*

$$\mathbb{E}[g_t \mid \theta_t] = \nabla L(\theta_t), \qquad \mathbb{E}\big[\|g_t - \nabla L(\theta_t)\|^2 \mid \theta_t\big] \, \leq \, \sigma^2 \tag{25}$$

*for some $\sigma^2 < \infty$.*

3. ***Robbins–Monro step sizes.*** *The learning rates $\{\eta_t\}_{t \geq 1}$ satisfy*

$$\sum_{t=1}^{\infty} \eta_t = \infty, \qquad \sum_{t=1}^{\infty} \eta_t^2 < \infty. \tag{26}$$

*Then the sequence $\{\theta_t\}$ produced by the ERPO update*

$$\theta_{t+1} = \theta_t - \eta_t g_t \tag{27}$$

*satisfies*

$$\lim_{t \to \infty} \mathbb{E}\big[\|\nabla L(\theta_t)\|^2\big] = 0, \tag{28}$$

*i.e., ERPO converges to a stationary point of the regularized loss $L(\theta)$ in the standard nonconvex SGD sense.*

*Proof sketch.* The proof follows the classical analysis of stochastic gradient descent for smooth nonconvex objectives. By $L$-smoothness of $L(\theta)$ and the ERPO update $\theta_{t+1} = \theta_t - \eta_t g_t$, we have

$$L(\theta_{t+1}) \, \leq \, L(\theta_t) - \eta_t \, \nabla L(\theta_t)^\top g_t + \frac{L}{2} \eta_t^2 \|g_t\|^2. \tag{29}$$

Taking conditional expectation with respect to $\theta_t$ and using the unbiasedness and bounded variance of $g_t$ yields

$$\mathbb{E}\big[L(\theta_{t+1}) \mid \theta_t\big] \, \leq \, L(\theta_t) - \eta_t \, \|\nabla L(\theta_t)\|^2 + \frac{L}{2} \eta_t^2 \big( \|\nabla L(\theta_t)\|^2 + \sigma^2 \big). \tag{30}$$

Rearranging terms and taking full expectation, one obtains a recursion of the form

$$\mathbb{E}\big[L(\theta_{t+1})\big] \;\leq\; \mathbb{E}\big[L(\theta_t)\big] - c_1 \eta_t \, \mathbb{E}\big[\|\nabla L(\theta_t)\|^2\big] + c_2 \eta_t^2, \tag{31}$$

for some constants $c_1, c_2 > 0$ depending on $L$ and $\sigma^2$. Summing this inequality over $t$ and using the Robbins–Monro conditions on $\{\eta_t\}$, along with the lower boundedness of $L(\theta)$, yields

$$\sum_{t=1}^{\infty} \eta_t \, \mathbb{E}\big[\|\nabla L(\theta_t)\|^2\big] < \infty. \tag{32}$$

This implies $\lim_{t\to\infty} \mathbb{E}[\|\nabla L(\theta_t)\|^2] = 0$, which completes the proof. $\qquad\square$

AN UPPER BOUND ON QUERY-KL (ENVIRONMENT DRIFT)

We finally show that, under a mild boundedness assumption on the query-weighted gain, the Query-KL term is uniformly bounded along the ERPO training trajectory, provided that the regularized loss does not increase beyond its initial value.

**Proposition 2** (Implicit upper bound on Query-KL). *Assume the query-weighted gain $\bar{g}_\theta(s)$ is uniformly bounded: there exists $G_{\max} > 0$ such that*

$$\big|\bar{g}_\theta(s)\big| \;\leq\; G_{\max} \quad \text{for all } \theta \text{ and } s. \tag{33}$$

*Consider the ERPO loss*

$$L(\theta) = -\mathbb{E}_{s\sim\rho_\theta}\big[\bar{g}_\theta(s)\big] + \lambda \, R_{query}(\theta), \qquad R_{query}(\theta) = \mathrm{KL}\big(\rho_\theta \,\|\, \rho_{\theta_0}\big). \tag{34}$$

*Assume that training is initialized at $\theta_0$ such that $R_{query}(\theta_0) = 0$, and that the ERPO updates satisfy*

$$L(\theta_t) \;\leq\; L(\theta_0) \quad \text{for all iterations } t. \tag{35}$$

*Then the Query-KL term is uniformly bounded:*

$$R_{query}(\theta_t) = \mathrm{KL}\big(\rho_{\theta_t} \,\|\, \rho_{\theta_0}\big) \;\leq\; \frac{L(\theta_0) + G_{\max}}{\lambda} \quad \text{for all } t. \tag{36}$$

*Proof.* By the boundedness of $\bar{g}_\theta(s)$, we have

$$-\mathbb{E}_{s\sim\rho_\theta}\big[\bar{g}_\theta(s)\big] \;\geq\; -G_{\max} \quad \text{for any } \theta. \tag{37}$$

Therefore,

$$L(\theta) = -\mathbb{E}_{s\sim\rho_\theta}\big[\bar{g}_\theta(s)\big] + \lambda \, R_{\text{query}}(\theta) \;\geq\; -G_{\max} + \lambda \, R_{\text{query}}(\theta). \tag{38}$$

For any iteration $t$, using the assumption $L(\theta_t) \leq L(\theta_0)$, we obtain

$$-G_{\max} + \lambda \, R_{\text{query}}(\theta_t) \;\leq\; L(\theta_t) \;\leq\; L(\theta_0). \tag{39}$$

Rearranging terms yields

$$R_{\text{query}}(\theta_t) \;\leq\; \frac{L(\theta_0) + G_{\max}}{\lambda}, \tag{40}$$

which shows that the Query-KL term, and hence the environment drift measured in KL divergence, is uniformly bounded throughout training. $\qquad\square$

## G  DERIVATION OF THE QUERY WEIGHTS

This appendix derives the closed-form batch weights $w_B(s)$ used in Section 4.2 from a self-normalized substitute distribution over queries.

**Self-normalized substitute distribution.** Consider the outer expectation over queries in the ERPO objective equation 6. On a mini-batch $B = \{s_i\}_{i=1}^m$ drawn from some proposal distribution (e.g., the current policy-induced query distribution $\rho_\theta$ or a replay buffer), we introduce a *batch self-normalized substitute distribution* over $B$:

$$\mu_B(s_i) \;=\; \frac{r_\theta(s_i)}{Z_B}, \qquad Z_B = \sum_{j=1}^m r_\theta(s_j), \tag{41}$$

where $r_\theta(s) > 0$ is a monotone score and $Z_B$ is its batch partition function. In the main text we choose $r_\theta(s) = 1/(-\ell_\theta(s))$ with $\ell_\theta(s) = \log P_\theta(s) < 0$ from equation 4, but the derivation below holds for any positive $r_\theta$.

Using $\mu_B$, we define the batch objective

$$J_B(\theta) \;:=\; \sum_{i=1}^m \mu_B(s_i)\, \bar{g}_\theta(s_i) \;=\; \frac{1}{Z_B} \sum_{i=1}^m r_\theta(s_i)\, \bar{g}_\theta(s_i), \tag{42}$$

where $\bar{g}_\theta(s)$ is the per-query expected return in equation 5. Crucially, $r_\theta$ and $Z_B$ are computed with **stop-gradient**: during backpropagation we treat them as constants and do not differentiate through the log-likelihoods used to form $r_\theta$. Under this convention, the gradient of $J_B$ is

$$\nabla_\theta J_B(\theta) \;=\; \frac{1}{Z_B} \sum_{i=1}^m r_\theta(s_i)\, \nabla_\theta \bar{g}_\theta(s_i). \tag{43}$$

Thus, relative to the unweighted estimator $\frac{1}{m} \sum_i \nabla_\theta \bar{g}_\theta(s_i)$, the query contributions are reweighted by $r_\theta(s_i)$, but the gradient remains a linear combination of per-query gradients.

**Scale-invariant surrogate.** The normalization constant $Z_B$ in equation 43 is independent of $\theta$, so it only affects the overall scale of the gradient, not its direction. We therefore introduce a scale-invariant surrogate

$$\widetilde{J}_B(\theta) \;:=\; \sum_{i=1}^m c_B\, r_\theta(s_i)\, \bar{g}_\theta(s_i), \tag{44}$$

where $c_B > 0$ is any batch-dependent constant that does not depend on $\theta$. Differentiating under the stop-gradient convention yields

$$\nabla_\theta \widetilde{J}_B(\theta) \;=\; c_B \sum_{i=1}^m r_\theta(s_i)\, \nabla_\theta \bar{g}_\theta(s_i) \;=\; c_B Z_B\, \nabla_\theta J_B(\theta). \tag{45}$$

Hence $\widetilde{J}_B$ and $J_B$ induce the same gradient direction, differing only by a positive scalar factor $c_B Z_B$. This allows us to replace the normalized weights $\mu_B$ by *unnormalized* but scale-adjusted weights $c_B r_\theta(s)$ without changing the SGD update direction.

**Closed-form query weights.** To obtain a concrete and numerically well-behaved choice of $c_B$, we use the batch-averaged log-likelihood

$$\bar{\ell}_B \;=\; \frac{1}{m} \sum_{j=1}^m \ell_\theta(s_j) \;<\; 0, \tag{46}$$

and set $c_B = -\bar{\ell}_B > 0$. With this choice and $r_\theta(s) = 1/(-\ell_\theta(s))$, the unnormalized weight for each query $s_i$ becomes

$$w_B(s_i) \;:=\; c_B\, r_\theta(s_i) \;=\; \frac{-\bar{\ell}_B}{-\ell_\theta(s_i)} \;=\; \frac{\bar{\ell}_B}{\ell_\theta(s_i)} \;>\; 0, \tag{47}$$

which is exactly the weight used in equation 8. Intuitively, this construction preserves the likelihood-based ordering of queries—higher likelihood (less negative $\ell_\theta(s)$) implies larger $r_\theta(s)$ and thus larger $w_B(s)$—while compressing the dynamic range compared to $\exp(\ell_\theta(s))$, avoiding the extreme ratios induced by log-normal importance weights.

Finally, substituting $w_B(s)$ into the surrogate objective $\widetilde{J}_B$ and normalizing by $m$ yields the query-reweighted Monte-Carlo estimator used in the main text,

$$\widehat{J}_B^{\text{reweight}}(\theta) \;=\; \frac{1}{m} \sum_{s \in B} w_B(s)\, \bar{g}_\theta(s), \tag{48}$$

which appears in equation 9 and constitutes the first term of the empirical ERPO objective equation 7. Under the assumptions in Appendix F, using this surrogate does not affect the convergence guarantee for SGD on the population objective $J_{\text{ERPO}}(\theta)$.

## H  USAGE OF LARGE LANGUAGE MODELS

In this work, we leveraged large language model to assist in the writing process by polishing the language. The LLM provided grammatical refinement, rephrased ambiguous expressions, and enhanced overall readability, while all technical content and claims remain the sole responsibility of the authors.

