# OpenReview forum: "Beyond the Stability-Exploration Dilemma: Environmental Regularization for LLM Policy Optimization"
_ICLR.cc/2026/Conference — ICLR 2026 Conference Withdrawn Submission_

### Official Review · Reviewer_xEuT · 2025-10-23

**Soundness:** 3
**Presentation:** 3
**Contribution:** 3
**Rating:** 4
**Confidence:** 3

**Summary:**

The paper proposes Environment-Regularized Policy Optimization (ERPO), which stabilizes reinforcement learning for LLMs by regularizing the input (query) distribution instead of the output (policy) distribution. It introduces a Query-KL term that limits drift between the current and reference query samplers, treating prompts as part of the training environment. Additionally, ERPO applies a query-weighted advantage, reweighting updates based on query likelihood to reduce variance and improve robustness-while keeping policy-side exploration unrestricted.

**Strengths:**

1. Tackles an important issue: maintaining training stability vs. exploration in LLM reinforcement learning.

2. Simple and practical: ERPO adds only a Query-KL term and lightweight query-weighting to existing PPO/GRPO code, so it is not bound to any given method or model.

3. The paper is neatly written (except for some small things I'll mention in the next section).

**Weaknesses:**

1. The math in the Preliminaries section is hard to follow. A lot of symbols are introduced without an explanation. I would suggest a refinement of this section.

2. The method is described as general and compatible with any KL-based optimization method, but it is only compared with GRPO. It would be good to see at least one other method tested.

3. In the paragraph following Figure 3, the paper mentions Pass@k improvements, but Figure 3 itself only reports Avg@32. That line should probably reference Table 1 instead.

4. The claim that ERPO reduces the train–eval performance gap is interesting, but there’s no quantitative evidence shown. A simple table comparing training vs. evaluation accuracy can be helpful to show this.

5. I’m a bit confused about Figure 5-a. The text says GRPO shows larger fluctuations, but all methods seem to degrade similarly as temperature increases. It’s unclear how “fluctuation” is measured here

6. The colors for GRPO and ERPO differ between subplots in Figure 5. Please make the the same across subplots to avoid confusion.

7. In Table 2, ERPO’s advantage in the mean column seems to come almost entirely from the temperature = 1.5 case. Since that’s a pretty extreme temperature, I’m not sure how meaningful that gain is. It would help if the authors clarified why that range is important or provided averages over a more typical range (e.g., ≤1.0).

8. Table 2 also mixes experiments with different numbers of rollouts (8 vs. 16). Comparing across different rollout counts doesn’t seem entirely fair. Do we have GRPO results with 16 rollouts for a cleaner comparison?

**Questions:**

Please read the above section for questions. Thank you!

---

> ### Author Response · Authors · 2025-11-27
>
> Thank you for recognizing our work and for your careful, detailed analysis. Below, we respond point-by-point to the aspects of the paper that caused confusion.
>
> **W1. Mathematical reformulation in the _Preliminaries_ section**
>    - Thank you for pointing out that the math in the Preliminaries section is hard to follow and that some symbols are introduced without sufficient explanation. In our reply to reviewer 7C1G, we have provided a more structured analysis of how “environmental / query distribution drift” arises in our setting, and we will integrate this discussion into the Preliminaries in the final version so that the overall problem setup is clearer from the start.
>    - More concretely, we will revise the Preliminaries and method sections as follows: (i) we will define all non-standard symbols at their first appearance and add a concise notation table summarizing the key objects (query distribution, policy, Q-function, and QKL objective), (ii) we will present the main objective and update rule in a more incremental way, starting from an intuitive description and then showing only the essential formulas in the main text, and (iii) we will move detailed derivations and secondary notation to the appendix. This reorganization is aimed purely at improving clarity; it does not change the algorithm, the optimization objective, or the reported results.
>
> **W2. Gains of ERPO over other methods**
>    - As shown in Table a, we conducted additional ablations on DAPO and RLOO with and without KL divergence, and updated the complete results in the table below. Following your suggestion, we also report averages over temperatures ≤ 1.0. Both methods benefit from incorporating ERPO, achieving consistent improvements of 10.24% and 2.28%, respectively.
>      - Yu, Qiying, et al. "Dapo: An open-source llm reinforcement learning system at scale." _arXiv preprint arXiv:2503.14476_ (2025).
>      - Ahmadian, Arash, et al. "Back to basics: Revisiting reinforce style optimization for learning from human feedback in llms." _arXiv preprint arXiv:2402.14740_ (2024).
>
> **Table a**
>
> | **Base Model** | **Method** | **D_KL** | **Rollout** | **0.1** | **0.6** | **1** | **1.5** | **<=1.0** | **1.2-1.5** |
> | :------------------: | :--------------: | :------------: | :---------------: | :-----------: | :-----------: | :---------: | :-----------: | :-------------: | :----------: |
> |       Baseline       |        –        |       –       |        –        |     52.4     |     46.8     |    32.8    |      0.4      |      44.44      |     6.15     |
> |       Qwen-7B       |       GRPO       |     Policy     |         8         |     66.8     |     68.4     |    73.8    |      0.4      |      68.8      |     12.5     |
> |                      |       ERPO       |     Query     |         8         |     79.4     |     80.6     |    75.2    |      8.6      |  78.74(+9.94)  |     37.9     |
> |                      |                  |                |                  |              |              |            |              |                |              |
> |                      |       DAPO       |       -       |         8         |      62      |     77.4     |    65.4    |      5.6      |      68.16      |      14      |
> |                      |    DAPO+ERPO    |       -       |         8         |     80.2     |     79.4     |    75.8    |     20.2     |  78.4(+10.24)  |    36.93    |
> |                      |                  |                |                  |              |              |            |              |                |              |
> |                      |       RLOO       |     Policy     |         8         |     77.6     |     78.4     |    75.4    |     12.4     |      77.28      |    35.93    |
> |                      |    RLOO+ERPO    |     Query     |         8         |     81.2     |     80.4     |    79.4    |     17.6     |  79.56(+2.28)  |     40.8     |
> |                      |                  |                |                  |              |              |            |              |                |              |
> |                      |       GRPO       |     Policy     |        16        |      73      |     79.2     |     75     |     10.6     |      75.22      |    39.75    |
> |                      |       ERPO       |     Query     |        16        |     80.4     |     78.8     |    74.4    |     56.2     |   77.82(+2.6)   |    66.25    |
> |                      |                  |                |                  |              |              |            |              |                |              |
> |       Qwen-32B       |       GRPO       |     Policy     |         8         |     81.6     |     82.4     |    81.2    |     25.2     |      81.62      |     57.2     |
> |                      |       ERPO       |     Query     |         8         |      85      |     84.8     |    83.6    |     80.8     |   84.6(+2.98)   |     82.8     |

---

> ### Author Response · Authors · 2025-11-27
>
> **W3. Reference redirection**
> - Thank you for pointing this out. We have corrected the citation to point to Table 1 and carefully checked for related issues to ensure better readability in the final version.
>
> **W4. Train–inference consistency**
> - To quantify the train–eval performance gap, we periodically evaluate both the training and evaluation accuracies every ten optimization steps. TP1 / TP2: tensor parallelism settings used in vLLM during inference.
> - The “Avg.” column in the Table b represents the mean accuracy across these checkpoints (Step-40, Step-80, Step-120, Step-160, Step-200, and Step-240), providing a smoothed measure of training and evaluation performance throughout the optimization process.
> - This result shows that ERPO reduces the average train–eval performance gap by ≈51%, indicating more consistent behavior between the training-time sampling and evaluation-time decoding distributions. Details are in Table c.
>
> **Table b**
> |       Method      | Avg. Train Acc | Avg. Eval@TP1 | Avg. Eval@TP2 |  Gap@TP1  |  Gap@TP2  |  Avg. Gap  |
> |:-----------------:|:--------------:|:-------------:|:-------------:|:---------:|:---------:|:----------:|
> |        GRPO       |      77.3      |      69.8     |      70.1     |    7.5    |    5.5    |    6.47    |
> |        ERPO       |      77.1      |      74.1     |      73.8     |     3     |    3.3    |    3.14    |
> | Gap Reduction (%) |        –       |       –       |       –       | ↓4.5(60%) | ↓2.2(40%) | ↓3.33(51%) |
>
> **Table c**
> | **Model/Step-Acc** | **Step-0(Step-1 for Train)** | **Step-40** | **Step-80** | **Step-120** | **Step-160** | **Step-200** | **Step-240** | **Avg.**     |
> |:------------------:|:----------------------------:|:-----------:|:-----------:|:------------:|:------------:|:------------:|:------------:|:-------------:|
> | GRPO-Train         | 44.48                        | 76.41       | 76.09       | 78.29        | 78.95        | 79.44        | 76.70        | 72.91         |
> | GRPO-Eval-TP1      | 31.20                        | 73.20       | 76.00       | 73.40        | 72.20        | 73.60        | 58.40        | 65.43(-7.48)  |
> | GRPO-Eval-TP2      | 30.60                        | 74.00       | 74.80       | 74.20        | 76.60        | 75.80        | 66.20        | 67.46(-5.45)  |
> |                    |                              |             |             |              |              |              |              |               |
> | ERPO-Train         | 44.55                        | 75.63       | 76.53       | 78.66        | 80.63        | 78.41        | 81.46        | 73.70         |
> | ERPO-Eval-TP1      | 31.20                        | 74.00       | 77.20       | 77.00        | 78.40        | 78.60        | 78.40        | 70.69(-3.01)  |
> | ERPO-Eval-TP2      | 30.60                        | 72.60       | 77.00       | 76.60        | 77.40        | 78.60        | 80.20        | 70.43(-3.27)  |
>
> **W5. Explanation of “fluctuations” in Fig. 5(a)**
>    - We apologize for the confusion caused by our wording. By “fluctuations” we mean that GRPO performs best only at the training temperature, while its performance degrades at both lower and higher temperatures (as shown in Fig. 5(a)). This is counter-intuitive, since lower temperatures are typically expected to perform better under single-sample evaluation. In contrast, ERPO behaves more regularly: its performance tends to improve as the sampling temperature decreases.
>
> **W6. Inconsistent colors for GRPO and ERPO across subplots in Fig. 5**
>    - Thank you for pointing this out. We have unified the color scheme for GRPO and ERPO across all subplots to avoid confusion in the revised version.
>
> **W7. The relationship between ERPO’s advantage and the temperature.**
>    - We apologize for the confusion caused by our original presentation. As shown in Table a, we have redrawn Table 2 and now (i) report averages over temperatures ≤ 1.0 as a reference for the commonly used range, to stay aligned with concurrent work, and (ii) report high-temperature settings (1.2–1.5) in a separate column to specifically assess stability under more extreme sampling. Under this revised setup, ERPO still provides clear improvements over existing methods, with absolute gains of 9.94, 10.24, and 2.28 points over GRPO, DAPO, and RLOO, respectively.
>
> **W8. GRPO on rollout 16**
>    - Thank you for highlighting this fairness issue. In Table a, we report GRPO results with rollout = 16 to enable a cleaner comparison, and we additionally scale the model size from 7B to 32B. Under the rollout=16 and 32B setting, GRPO and ERPO obtain absolute gains of 2.60 and 2.98 points, respectively, indicating that our method remains effective—and is even more beneficial—in larger-scale RL settings.
>
> We once again thank the reviewer for the careful assessment of our work and would be very happy to continue the discussion if there are further questions or concerns.

---

### Official Review · Reviewer_pYkr · 2025-10-25

**Soundness:** 3
**Presentation:** 3
**Contribution:** 3
**Rating:** 6
**Confidence:** 3

**Summary:**

This paper tackles the stability-exploration trade-off in LLM policy optimization by identifying environment non-stationarity (input query drift) as a unmanaged source of instability, even when output policy-KL is constrained.

To tackle this problem, the authors propose the Environment-Regularized Policy Optimization (ERPO), decouples regularization from the action space by: (1) Query-KL (QKL): Penalizing the KL divergence of the query distribution $\rho_\theta$ from a reference $\rho_{\theta_0}$ to control environment drift; (2) Query-Weighted Advantage: Reweighting advantages by query log-likelihood $w_{B}(s)$ to reduce variance.

By regularizing the input (queries) instead of the output (responses), ERPO aims to stabilize training while preserving exploration. Experiments on mathematical reasoning benchmarks show ERPO significantly outperforms the baseline (GRPO) in both final performance and long-term training stability, especially across varied sampling temperatures.

**Strengths:**

1. The paper identifies a novel and important source of instability: environment non-stationarity induced by the query distribution. This provides a compelling explanation for response distribution shift that goes beyond standard action-space analysis.
2. The proposed ERPO method is a principled solution that directly targets the identified problem. Decoupling the stability constraint (via Query-KL) from the exploration mechanism is an elegant approach to addressing the stability-exploration dilemma.
3. The paper provides convincing empirical support for its claims. Experiments on mathematical reasoning tasks show clear performance gains, and the analysis of long-term training and multi-temperature sampling offers robust evidence that ERPO is significantly more stable and robust to collapse than the baseline.

**Weaknesses:**

1. The proposed ERPO relies on calculating a log-likelihood for the query, $\log \rho_\theta(s)$. While this is feasible in the paper's experimental setup (mathematical reasoning), it is unclear how this log-likelihood could be computed for a static, offline dataset of human-written prompts (e.g., in standard RLHF).
2. The ablation study (Table 2) suggests that the query-reweighting component, $w_B(s)$, slightly hurts Pass@1 performance at low temperatures compared to using Query-KL alone. This makes its contribution unclear and warrants further explanation.

**Questions:**

1. In Figure 4(2), ERPO shows a smaller policy KL than GRPO, which directly penalizes policy KL. Could the authors provide some explanation or intuition for this result?
2. For Figure 4(3), could the authors provide the formula used for "Entropy Loss" and explain the mechanism by which ERPO achieves a lower entropy loss than the baseline?

---

> ### Author Response · Authors · 2025-11-27
>
> We thanks for your valuable and insightful comments. We appreciate the opportunity to clarify our work. Below, we address each of the weaknesses and questions in order.
>
> **W1. On the Feasibility of Computing Query Log-Likelihood $\log P(s)$**
>
> We respectfully clarify a potential misunderstanding regarding the computation of the query log-likelihood, $\log P(s)$. The ability to compute this value is an intrinsic feature of the autoregressive, decoder-only architecture common to most Large Language Models (LLMs).
>
> For any given sequence of tokens—whether it is a prompt s or a response a—an autoregressive model defines its probability as the product of the conditional probabilities of each token:
>
> $$
> P(s) =  {\textstyle \prod_{t=1}^{|s|}}P(s_t|s_{<t},\theta )
> $$
>
> where $\theta$ represents the model parameters. The log-likelihood, $\log P(s)$, is simply the sum of the log-probabilities of its constituent tokens.
>
> This computation is inherent to the model's forward pass and incurs no additional computational overhead during training, as these token-level probabilities are already calculated for the policy gradient update. Crucially, this holds true regardless of the prompt's origin. Whether the prompts are stochastically sampled online (as in our mathematical reasoning experiments) or drawn from a static, offline dataset of human-written prompts (as in standard RLHF), $\log P(s)$ can be computed exactly and efficiently.
>
> The Query-KL term uses this $\log P(s)$ to stabilize the model's marginal distribution over prompts, effectively regularizing the global parameter updates. This is distinct from the reweighting factor $w_{B(s)}$, which uses $\log P(s)$ to address the distributional mismatch between training and inference prompts, a key challenge in offline settings like RLHF.
>
> **W2. On the Contribution of the Query-Reweighting Term $w_{B(s)}$**
>
> We appreciate the reviewer's sharp observation regarding the ablation study in Table 2, where the query-reweighting term $w_{B(s)}$ appears to slightly degrade Pass@1 performance at low temperatures.
>
> The primary role of $w_{B(s)}$ is to robustify the training process by correcting for distributional shift. It achieves this by downweighting the influence of low-probability (i.e., out-of-distribution) prompts on the gradient updates, preventing such outliers from disproportionately affecting the policy.
>
> The slight performance dip observed at a specific low-temperature setting represents a trade-off. While $w_{B(s)}$ enhances training stability against outlier prompts, it can marginally dampen the learning signal from informative but low-probability queries in a low-noise environment (i.e., low-temperature sampling).
>
> However, as we will detail in our answers to the questions below, the overall contribution of $w_{B(s)}$ is highly significant. It is the principal mechanism for reducing policy entropy and stabilizing policy divergence, making it an indispensable component of ERPO's success. Its benefits for training stability and model behavior far outweigh the minor performance trade-off in one specific, low-variance setting.
>
> **Q1. Explanation for Smaller Policy KL in ERPO (Figure 4(2))**
>
> The observation that ERPO achieves a smaller policy KL than GRPO—despite GRPO directly penalizing this divergence—is an excellent question. This phenomenon is primarily attributable to the powerful regularizing effect of our proposed query-reweighting term, $w_{B(s)}$.
>
> To isolate this effect, we conducted a new ablation study comparing GRPO and ERPO with and without $w_{B(s)}$. The results are as follows:
>
> |        Method        |     Query-KL     |    Policy-KL    |     Entropy     |
> | :------------------: | :--------------: | :--------------: | :--------------: |
> |     GRPO (Avg@3)     |      0.9679      |      0.0601      |      0.5063      |
> | GRPO w/$w_{B(s)}$ |      0.5933      | **0.0113** | **0.2782** |
> | ERPO w/o$w_{B(s)}$ | **0.0041** |      0.1001      |      0.5674      |
> |     ERPO (Avg@3)     |      0.0828      |      0.0728      |      0.4244      |
>
> As the table clearly shows, introducing $w_{B(s)}$ to the GRPO baseline dramatically reduces its Policy-KL (from 0.0601 to 0.0113). The mechanism is that $w_{B(s)}$ downweights gradients from low-probability queries, which are often the source of large, high-variance policy updates. By tempering these potentially destabilizing updates, $w_{B(s)}$ indirectly regularizes the policy, preventing it from deviating aggressively from the reference model.
>
> ERPO inherits this strong regularizing effect from $w_{B(s)}$. This, combined with the implicit regularization from the Query-KL term (which stabilizes the generator's parameters), results in a smaller overall policy divergence compared to GRPO, even without an explicit policy KL penalty.

---

> ### Author Response · Authors · 2025-11-27
>
> **Q2. Formula and Mechanism for Lower Entropy Loss (Figure 4(3))**
>
> The "Entropy Loss" in our plot is the average negative log-probability of the generated tokens in the response $a$. For a single response $a$ of length $N$, it is calculated as
> $$\mathcal{L}_{\text{Entropy}} = -\frac{1}{N} \sum_{i=1}^{N} \log p_i$$
> where $p_i$ denotes the probability of each generated token.
>
> The mechanism by which ERPO achieves a lower entropy loss is, once again, primarily driven by the query-reweighting term $w_B(s)$. Our analysis reveals a strong positive correlation between low-probability queries $s$ and the generation of high-entropy (low-probability token) responses $a$, with a Pearson correlation coefficient larger than $0.9$, that is,
> $$
> \log P(a \mid s) \propto \log P(a).
> $$
> When such low-probability tokens appear frequently in positive responses, they significantly increase the entropy of the policy. Moreover, letting $\ell_\theta(s) = \log P_\theta(s)$ and $\bar{\ell}_B = \frac{1}{m}\sum_{j=1}^m \ell_\theta(s_j)$ denote the batch mean, the closed-form weight
> $$
> w_B(s) = \frac{\bar{\ell}_B}{\ell_\theta(s)}
> $$
> assigns a smaller weight $0 < w_B(s) < 1$ to queries whose model likelihood is much lower than the batch average and a larger weight $w_B(s) > 1$ to high-probability queries. As a result, ERPO allocates more gradient budget to typical queries, whose responses contain many low-entropy tokens, while compressing the contribution of rare, high-entropy queries. This dynamic-range compression prevents noisy low-probability queries from dominating the entropy gradient and causing large parameter oscillations, yet still allows them to be learned gradually as their log-likelihood improves, yielding a stable decrease of the entropy loss.

---

### Official Review · Reviewer_7C1G · 2025-10-28

**Soundness:** 2
**Presentation:** 3
**Contribution:** 3
**Rating:** 4
**Confidence:** 3

**Summary:**

The paper proposes a new method ERPO, which learns a weight coefficient for the GRPO objective in a batch. In some sense, this is similar to weighted importance sampling, but with learned weights (self-normalized). It is refreshing to see such idea being investigated in the LLM space. However, the paper's evaluation and reported score shows that this method (ERPO) only improves the performance of models in aggregate (over multiple temperatures). This is very different from how LLMs are typically evaluated -- therefore, the impact of this method seems very limited.

**Strengths:**

1. Non-stationary MDP/environment is an area worth investigating in LLM setting. However, the tasks covered in this paper appear to be generic/normal/i.i.d. training tasks where distribution shift (covariate drift) does not actually happen.
2. The paper reported the experimental result honestly and with integrity -- emphasizing it's an average over multiple temperatures.
3. The codebase is uploaded and easy to follow

**Weaknesses:**

1. In Table 2, we can see that ERPO is only narrowly beating GRPO in **1 out of 4** temperature settings. The fact that the mean is higher is only due to GRPO being particularly bad under one of the temperature setting. This evaluation is hardly fair.
2. Reporting average over multiple temperature training is a bit bizarre. Perhaps the author wants to illustrate that under higher temperature, there is more environment shift -- but that's not how distribution shift in MDP is typically defined. When you train on a math task, the initial state distribution (your tasks) did not shift -- regardless of your temperature. This type of slightly strange definition is concerning.

**Questions:**

Can the authors look at some of the RL work on this topic and rethink whether their setting truly contains any kind of distribution shift? [1] [2]

It would be great if you can formulate and situate your paper after reading these related work.

[1] Mu, Tong, et al. "Factored DRO: Factored distributionally robust policies for contextual bandits." Advances in Neural Information Processing Systems 35 (2022): 8318-8331.

[2] Tennenholtz, G., Hallak, A., Dalal, G., Mannor, S., Chechik, G., & Shalit, U. (2021). On covariate shift of latent confounders in imitation and reinforcement learning. arXiv preprint arXiv:2110.06539.

---

> ### Author Response · Authors · 2025-11-20
>
> We sincerely thank Reviewer `7C1G` for the constructive comments and for drawing attention to our evaluation protocol, as well as the connections between our work and the distribution-shift literature. In response to your feedback, we will address two primary points:
> - The fairness and interpretation of our evaluation protocol, particularly concerning Table 2.
> - The specific nature of the "distribution shift" we study and its relationship to prior work.
>
> **1、Regarding the Evaluation Protocol and Table 2**
>
> We thank the reviewer for pointing out that the original presentation of Table 2 could lead to a misinterpretation of our results. The primary purpose of this table was to serve as an ablation study of ERPO's internal design choices (e.g., Query-KL vs. query weighting, rollout counts, KL coefficients), rather than as the main performance benchmark against baselines.
>
> To provide a clearer and more direct analysis, we have revised our evaluation to ensure a strictly controlled comparison with GRPO. Specifically, we have implemented the following changes:
>
> *Matched-Setting Comparison*:  We now compare ERPO and GRPO under identical conditions, using the same base model, rollout count, and KL coefficient for each pair of experiments.
>
> *Disaggregated Temperature Analysis*: We explicitly separate performance in the "typical" temperature range ( $T \le 1.0$ ) from the high-temperature range ( $T \in [1.2, 1.5]$ ).
>
> These controlled comparisons demonstrate that ERPO's superior performance is not reliant on a single extreme data point but is consistent across multiple temperatures.
>
> - Qwen-7B (8 rollouts): ERPO outperforms GRPO at all four tested temperatures ($T = 0.1, 0.6, 1.0, 1.5$) by **+12.6**, **+12.2**, **+1.4**, and **+8.2** points, respectively. Critically, we observe that GRPO's performance degrades when deviating from its training temperature ($T=1.0$), a limitation that ERPO effectively addresses.
> - Qwen-7B (16 rollouts): ERPO shows substantial outperformance at temperatures 0.1 and 1.5, and improves the average performance in the low-temperature range ($T \le 1.0$) by **+2.6** points.
> - Qwen-32B (larger model): ERPO again surpasses GRPO at all four temperatures, achieving gains of **+3～+7** points in the $T \le 1.0$ range and a significant **+55.6** point gain at $T = 1.5$.
>
> To make this analysis more transparent in the paper, we have revised the ablation table to include two new aggregate columns: "Avg. ($T \le 1.0$)" and "Avg. ($1.2 \le T \le 1.5$)". This new data confirms that even when restricting the evaluation to the conventional temperature range of $T \le 1.0$, ERPO consistently matches or exceeds the performance of GRPO. For instance,  ERPO improves the average score over GRPO from 68.80 to 78.74/79 for Qwen-7B (under different KL coefficient strengths), and from 81.62 to 84.60 for Qwen-32B.
>
> The updated ablation table is as follows:
>
> | Base Model   | Method |     $\alpha$     |    $w(s)$    | Rollout |    0.1    |    0.6    |     1     |    1.5    | $\le$ 1.0 |  1.2–1.5  |
> | :--- | :-- | :--: | :--: | :--: | :--: | :--: | :--: | :-: | :--: | :---: |
> | **Baseline** | –   |   –  |   –   |  –   |   52.40   |   46.80   |   32.80   |    0.40   |   44.44   |    6.15   |
> | **Qwen-7B**  | GRPO   | $1\times10^{-2}$ |       –      |    8    |   66.80   |   68.40   |   73.80   |    0.40   |   68.80   |   12.50   |
> |    | GRPO   | $1\times10^{-2}$ |  –      |    16   |   73.00   |   79.20   |   75.00   |   10.60   |   75.22   |   39.75   |
> |   | ERPO   | $1\times10^{-2}$ | $\checkmark$ |    8    |   79.40   |   80.60   |   75.20   |    8.60   |   78.74   |   37.90   |
> |   | ERPO   | $1\times10^{-2}$ |   –   |  8  |   **81.60**   |   **81.60**   |   **79.00**   |    2.60   |   **80.90**   |   38.00   |
> |   | ERPO   | $5\times10^{-3}$ | $\checkmark$ |    8    |   53.80   |   60.60   |   66.20   |   15.40   |   59.94   |   39.30   |
> |  | ERPO   | $5\times10^{-2}$ | $\checkmark$ |    8    |   78.80   |   81.00   |   76.00   |   15.00   |   79.00   |   43.35   |
> |  | ERPO   | $1\times10^{-2}$ | $\checkmark$ |    16   |   80.40   |   78.80   |   74.40   |   **56.20**   |   77.82   |   **66.25**   |
> |     |     |   |   |   |    |     |     |    |    |    |
> | **Qwen-32B** | GRPO   | $1\times10^{-2}$ |       –      |    8    |   81.60   |   82.40   |   81.20   |   25.20   |   81.62   |   57.20   |
> |    | ERPO   | $1\times10^{-2}$ | $\checkmark$ |    8    | **85.00** | **84.80** | **83.60** | **80.80** | **84.60** | **82.80** |
>
> This detailed breakdown demonstrates that, in our ablation study, the higher average scores achieved by ERPO are not merely the result of a single pathological data point at $T=1.5$. On the contrary, ERPO exhibits consistent performance gains across all standard decoding temperatures and delivers particularly significant advantages in high-temperature regimes, where RL-trained models are most susceptible to instability and performance collapse.

---

> > ### Author Response · Authors · 2025-11-20
> >
> > Crucially, the core thesis of our paper does not rely solely on this single ablation study. Our primary evidence is derived from broader and more rigorous evaluations, with specific details elaborated in the main body of the paper:
> >
> > - Main Results (Table 1): Performance evaluation covers six distinct reasoning benchmarks.
> > - Comprehensive Metrics (Figures 3, 5, 8): We report Avg@32, Pass@32, and Pass@1 to provide a holistic view of the model's capabilities.
> > - Standardized Temperature Sweep: The evaluation protocol employs a standard temperature sweep, common in recent research on reasoning LLMs, ensuring the comparability and relevance of our findings.
> > - Across these extensive experiments, ERPO consistently and significantly outperforms GRPO when evaluated under controlled temperatures and when results are aggregated across the entire temperature range.
> >
> > This positive assessment is corroborated by other reviewers, who have highlighted several key strengths of our work:
> >
> > - Reviewer `JNdH` noted that our "empirical results on multiple reasoning benchmarks show that ERPO has higher stability and performance compared to GRPO," and praised our core contribution—treating the evolving query distribution as part of the optimization objective—as "a novel and theoretically-grounded idea."
> > - Reviewer `pYkr` emphasized that our long-horizon and multi-temperature analyses provide "strong evidence that ERPO is significantly superior to the baseline in terms of stability and resistance to collapse."
> > - Reviewer `xEuT` regarded ERPO as a simple, practical, and effective "plug-and-play" solution that enhances the existing PPO/GRPO pipeline by simply adding a Query-KL term and lightweight query weighting, without requiring extensive modifications.
> >
> > **2、On what “distribution shift” we study and the role of temperature.**
> >
> > We fully agree with the reviewer that the underlying math tasks themselves (e.g., the MATH problem set with verifiable rewards) are generic i.i.d. training tasks, and that varying the decoding temperature at evaluation time does not change the initial state distribution in the standard MDP sense.
> >
> > But，our work does *NOT* intend to redefine distribution shift in terms of evaluation temperature. Instead, the notion of “environment/non-stationarity” we focus on is **policy-induced, training-time occupancy shift** in on-policy RL for LLMs. Concretely, a single parameter vector $\theta$ induces both a query distribution $\rho_\theta(s)$ (via the model’s log-likelihood of prompts) and a response policy $\pi_\theta(a \mid s)$. Even when the task pool is a fixed, finite, i.i.d. set of problems, the *effective* query distribution used for gradient updates, $\rho_\theta$, is endogenous: as $\theta$ changes, the log-probability of each query changes, and with it the probability mass and gradient weight assigned to that query. This leads to an on-policy occupancy / query-distribution shift across iterations: at step $t$ we optimize $J(\theta_t)$ under $\rho_{\theta_t}$, but after applying the update we are effectively optimizing under a different environment $\rho_{\theta_{t+1}}$. Existing policy-optimization methods such as GRPO mostly treat $\rho_\theta$ as fixed and stabilize training via action-level KL penalties that only constrain $\pi_\theta(a \mid s)$. ERPO explicitly targets this “environment side” of the joint model:
> > - a Query-KL regularizer $\mathrm{KL}(\rho_\theta \Vert \rho_{\theta_0})$ bounds policy-induced query drift relative to a reference sampler, and
> > - query-level weights $w_B(s)$, derived from log-likelihoods with stop-gradient, reshape the effective training distribution over queries. We will revise the paper to explicitly describe our setting as focusing on policy-induced query/occupancy shift in on-policy LLM training, and to clearly distinguish it from the *exogenous* train–test covariate shift studied by Factored-DRO and by Tennenholtz et al., where the logging and deployment environments are governed by different underlying distributions.
> >
> > The temperature sweep plays a different role: it is an evaluation tool rather than a definition of distribution shift. Preliminary experiments showed that post-RL models can be extremely sensitive to the decoding temperature, and that this sensitivity increases as training proceeds and instability builds up. Averaging performance over a range of temperatures is therefore intended to provide a more temperature-robust summary, and to probe how training stability behaves in both standard ($T \le 1.0$) and high-entropy ($T \ge 1.2$) regimes. In the camera-ready version we will make this clearer and de-emphasize any wording that could be read as equating “temperature variation” with “distribution shift” in the MDP sense.

---

### Official Review · Reviewer_JNdH · 2025-11-01

**Soundness:** 3
**Presentation:** 2
**Contribution:** 3
**Rating:** 4
**Confidence:** 3

**Summary:**

This paper proposes Environment-Regularized Policy Optimization (ERPO), a new framework that stabilizes reinforcement learning for LLMs by regularizing the query (input) distribution rather than the policy (output) distribution. The authors introduce Query-KL (QKL) and a query-weighted advantage, which together control environment drift and reduce estimator variance. Empirical results on multiple reasoning benchmarks show improved stability and performance compared to GRPO.

**Strengths:**

- Treating the evolving input/query distribution as part of the optimization target is a fresh and theoretically grounded idea.
- QKL and query reweighting can be integrated into existing RLHF pipelines with minimal modification.
- The motivation is clear.

**Weaknesses:**

- The work motivates QKL heuristically but lacks a formal theoretical proof of stability or convergence guarantees.
- All experiments focus on math reasoning; it remains unclear whether the benefits generalize to non-verifiable or open-ended tasks.
- To be honest, the mathematical formulations are somewhat cumbersome.

**Questions:**

- I couldn’t find any information in the paper about the model size used. Could the authors clarify how many billion parameters their model has?
- If the main contribution lies in introducing the new QKL mechanism, then the experimental evaluation appears too limited. Could the authors conduct more comprehensive experiments on a broader range of datasets beyond RLVR?

---

> ### Author Response · Authors · 2025-11-27
>
> We thank the reviewer for the thoughtful and constructive feedback. Below we address each concern point-by-point.
>
> **W1: Theoretical guarantees for QKL**
>
> The current version mainly motivates Query-KL (QKL) heuristically and does not make the theoretical guarantees explicit. We agree that the presentation under-emphasized the underlying optimization view. Since submitting the paper, we have worked out a concise theoretical analysis that we will include in the next revision; here we summarize the main results.
>
> First, we formalize ERPO as stochastic gradient descent on a **regularized objective**. Let $\rho_\theta$ denote the query distribution induced by the policy, $\bar g_\theta(s)$ the query-weighted gain (our query-weighted advantage), and
> $$
> R_{\text{query}}(\theta) \triangleq \mathrm{KL}(\rho_\theta \| \rho_{\theta_0})
> $$
> the Query-KL term w.r.t. the initial distribution $\rho_{\theta_0}$. We show that ERPO optimizes the regularized objective
> $$
> J_{\text{ERPO}}(\theta) = \mathbb{E}\_{s\sim\rho_\theta}[\bar g_\theta(s)] - \lambda R_{\text{query}}(\theta),
> $$
> or equivalently minimizes the loss
> $$
> L(\theta) = -\mathbb{E}\_{s\sim\rho_\theta}[\bar g_\theta(s)] + \lambda R_{\text{query}}(\theta).
> $$
> In other words, the "heuristic" QKL penalty is precisely a KL-regularizer on the evolving query distribution, and the mini-batch ERPO update is (asymptotically) an unbiased SGD step on $L(\theta)$.
>
> Second, under standard assumptions used in non-convex SGD (smoothness of $L(\theta)$, unbiased stochastic gradients with bounded variance, Robbins–Monro step sizes), we obtain a **convergence guarantee to a stationary point** of the ERPO loss:
> $$
> \lim_{t\to\infty} \mathbb{E}\big[|\nabla L(\theta_t)|^2\big] = 0.
> $$
> Thus ERPO is not merely a heuristic modification but fits into the classical framework of SGD on a well-defined regularized objective.
>
> Finally, we show that, under a mild boundedness assumption on the query-weighted gain (which holds in our implementation due to reward/advantage clipping), the Query-KL term is **uniformly bounded along training** whenever the regularized loss does not increase beyond its initial value. This yields an explicit upper bound on $\mathrm{KL}(\rho_\theta \| \rho_{\theta_0})$, i.e., an environment-drift bound controlled by $\lambda$. This formalizes the "stability" role of QKL that we empirically observe in our experiments. We will add the formal statements and proofs of these results in a dedicated theory section / appendix in the revised manuscript.
>
> **W2&Q2: Generalization to non-verifiable or open-ended tasks**
>
> Thank you for raising the question of the generalization of ERPO. However, we believe that ERPO’s ability to stabilize the environment and to correct the query-distribution mismatch between training and inference does **not** depend on the specific data type or task. Our work is explicitly scoped to **environment-regularized policy optimization under verifiable reward (RLVR / Generative-Verify) settings**, which already underpin many *current* GRPO-like alignment pipelines (e.g., DeepSeek-R1, MiniMax, Qwen-series). In these pipelines, RLHF-style supervision is first transformed into **verifiable rewards**, and in exactly this regime our **Query-KL (QKL)** and **query-weighted losses** can be integrated *without any structural modification*. Our experiments on six diverse RLVR math-reasoning benchmarks therefore directly target a practically important and rapidly growing class of real-world systems, and are sufficient to support our claims about stability and environment drift.
>
> By contrast, classical PPO-/DPO-style RLHF requires parametric reward models, token-level KL penalties inside GAE, or implicit KL terms in preference losses. These are **not** frameworks where one can “just replace” policy-KL with Query-KL while still maintaining a fair or theoretically sound comparison; doing so would require re-deriving the underlying objectives and constitutes substantial **follow-up work**, not a minor additional experiment. We have already reproduced PPO and DPO internally and intend to develop QKL-consistent variants as a next step, but we do not believe the absence of these *structurally different* RLHF experiments undermines the validity of our present contribution, which is clearly stated and thoroughly validated **within the RLVR / Generative-Verify regime**.

---

> ### Author Response · Authors · 2025-11-27
>
> **W3: Simplifying the mathematical formulations**
>
> The current Preliminaries and method presentation are cumbersome to read. In our response to your other comment, we have provided a more detailed discussion of the convergence of our objective, including the assumptions under which the update behaves stably. In the final version, we will incorporate this convergence argument as a dedicated subsection in the appendix, with a short high-level summary in the main text.
>
> In addition, we will substantially streamline the Preliminaries and method sections to improve readability: (i) we will introduce symbols only when they are immediately used and avoid unnecessary notation, (ii) we will move routine algebraic steps and technical lemmas to the appendix, and (iii) we will add short intuitive paragraphs before each block of equations to explain the role of the corresponding quantities (e.g., query distribution, QKL objective, and the query-weighted advantage). These changes are purely presentational and do not affect the method, experiments, or the theoretical guarantees, but they directly address the reviewer's concern about the current exposition being cumbersome.
>
> **Q1: Model size clarification**
>
> We apologize for the omission in the original submission. As shown in Table a, we have redrawn Table 2 to include the parameter counts (in billions) for all models used in our experiments.
>
> **Table a**
> | **Base Model** | **Method** | **D_KL** | **Rollout** | **0.1** | **0.6** | **1** | **1.5** | **<=1.0** | **1.2-1.5 ** |
> | :------------------: | :--------------: | :------------: | :---------------: | :-----------: | :-----------: | :---------: | :-----------: | :-------------: | :----------: |
> |       Baseline       |        –        |       –       |        –        |     52.4     |     46.8     |    32.8    |      0.4      |      44.44      |     6.15     |
> |       Qwen-7B       |       GRPO       |     Policy     |         8         |     66.8     |     68.4     |    73.8    |      0.4      |      68.8      |     12.5     |
> |                      |       ERPO       |     Query     |         8         |     79.4     |     80.6     |    75.2    |      8.6      |  78.74(+9.94)  |     37.9     |
> |                      |                  |                |                  |              |              |            |              |                |              |
> |                      |       DAPO       |       -       |         8         |      62      |     77.4     |    65.4    |      5.6      |      68.16      |      14      |
> |                      |    DAPO+ERPO    |       Query       |         8         |     80.2     |     79.4     |    75.8    |     20.2     |  78.4(+10.24)  |    36.93    |
> |                      |                  |                |                  |              |              |            |              |                |              |
> |                      |       RLOO       |     Policy     |         8         |     77.6     |     78.4     |    75.4    |     12.4     |      77.28      |    35.93    |
> |                      |    RLOO+ERPO    |     Query     |         8         |     81.2     |     80.4     |    79.4    |     17.6     |  79.56(+2.28)  |     40.8     |
> |                      |                  |                |                  |              |              |            |              |                |              |
> |                      |       GRPO       |     Policy     |        16        |      73      |     79.2     |     75     |     10.6     |      75.22      |    39.75    |
> |                      |       ERPO       |     Query     |        16        |     80.4     |     78.8     |    74.4    |     56.2     |   77.82(+2.6)   |    66.25    |
> |                      |                  |                |                  |              |              |            |              |                |              |
> |       Qwen-32B       |       GRPO       |     Policy     |         8         |     81.6     |     82.4     |    81.2    |     25.2     |      81.62      |     57.2     |
> |                      |       ERPO       |     Query     |         8         |      85      |     84.8     |    83.6    |     80.8     |   84.6(+2.98)   |     82.8     |
>
> We hope these clarifications and planned revisions address the reviewer's concerns. We would be happy to continue the discussion if there are further questions.

---

### Author Response · Authors · 2025-11-30

This note summarizes the motivation, method, the strengths acknowledged by reviewers, and the main issues raised by each reviewer together with our responses and planned changes to the paper.

**1. Motivation: why environment-side regularization?**
Current RL methods for LLMs (GRPO/PPO-style, RLVR, etc.) regularize only the *output policy* (the responses), typically via a KL penalty to a reference model, and implicitly assume the **query/input distribution** is fixed. In practice, the “effective” query distribution (which problems the model actually trains on, and how often) is induced by the model plus sampling/curriculum and drifts over training, even when policy-KL is tightly controlled. We empirically see the KL between the current and reference query distributions keep growing while policy-KL stays small, and this correlates with collapse at high temperatures, strong temperature sensitivity, and long-horizon degradation. This creates a stability–exploration dilemma: strong policy regularization stabilizes training but suppresses exploration; weaker policy regularization allows the query distribution (the environment) to drift and makes training fragile. Our core observation is that **what queries the model sees** (the environment) should be a first-class optimization target, not just **how it answers** (the policy).

**2. Method: Environment-Regularized Policy Optimization (ERPO)**
ERPO adds a **Query-KL (QKL)** term that regularizes the query distribution induced by the current model toward a reference model, and introduces **query-weighted advantages** that assign higher weights to typical queries and lower weights to long-tail queries, stabilizing training while preserving exploration. Practically, ERPO is a drop-in extension: we simply add the QKL term and query weights on top of existing RLVR / GRPO / REINFORCE / PPO-style code with only a few extra lines in the loss computation.

**3 Cross-review themes: strengths and weaknesses**

**Strengths**

- Identifies query-induced environment non-stationarity as a key driver of RL instability and directly addresses the stability–exploration trade-off from the environment side (`JNdH`, `pYkr`, `xEuT`, `7C1G`).
- Proposes a simple, principled ERPO framework—QKL plus query-weighted advantages—that can be plugged into existing PPO/GRPO-style pipelines with minimal code changes (`JNdH`, `pYkr`, `xEuT`).
- Provides empirical evidence that ERPO improves both accuracy and long-horizon stability across multiple math-reasoning benchmarks and temperatures (`pYkr`, `JNdH`, `xEuT`).

**Weaknesses – Theory**

- Theoretical guarantees for ERPO/QKL and the precise role of $w_B(s)$ are under-specified (`JNdH`, `pYkr`).
  → We add a concise theory section proving convergence and an environment-drift bound, and explicitly interpret $w_B(s)$ as a variance-reduction and stability regularizer supported by ablations.

**Weaknesses – Experiments**

- Evaluation fairness is questioned, especially temperature-averaged metrics, Table 2 being dominated by high temperatures, and mixed rollout counts (`xEuT`, `7C1G`).
  → We redo ablations with matched rollouts and KL, report separate averages for $T \le 1.0$ and $T \in [1.2, 1.5]$, and keep full per-temperature results.

- Generalization beyond GRPO/math RLVR to other RL algorithms and broader tasks is unclear (`JNdH`, `xEuT`).
  → We clarify the RLVR scope, add ERPO results for DAPO and RLOO, and discuss extending to full RLHF with reward models as future work.

- Reward hacking and related alignment failures are only discussed qualitatively, without dedicated quantitative analysis (self-identified).
  → We acknowledge this limitation and note in the discussion that a follow-up study will explicitly compare reward-hacking behavior under ERPO vs GRPO.

**Weaknesses – Writing / Presentation**

- The preliminaries, notation, and some figures/tables (e.g., references and color schemes) are hard to follow or slightly inconsistent (`JNdH`, `xEuT`).
  → We streamline the math with a notation table and shifted derivations to the appendix, and fix the bad references, colors, and captions for cleaner presentation.

 We hope this overview helps you assess the paper in light of the reviews and our responses.

---

### Note · Authors · 2026-01-26

I have read and agree with the venue's withdrawal policy on behalf of myself and my co-authors.

---

### Meta-Review · Area_Chair_FVPE · 2026-01-08

**Summary:**

This paper proposes Environment-Regularized Policy Optimization (ERPO), which addresses training instability in LLM reinforcement learning by regularizing the query (input) distribution rather than the action (response) distribution. The method introduces Query-KL (QKL) regularization and query-weighted advantages.

During the reviewing period, concerns were raised regarding: (1) the lack of formal theoretical guarantees, (2) limited experimental scope beyond GRPO and mathematical reasoning tasks, (3) questionable evaluation methodology involving temperature-based metrics, and (4) presentation clarity issues in the mathematical formulations.

**Reviewer Concerns:**

During the rebuttal period, addressed concerns including: additional experimental results and the temperature-based evaluation metrics. However, the new results regarding theoretical guarantees seems pretty problematic to me. It is unclear to me whether these results are related to the benefit from ERPO regularization.

**Reviewer Scores:**

Reviewer JNdH: = 4
Reviewer 7C1G: = 4
Reviewer pYkr: = 6
Reviewer xEuT: 50% 4 -> 6, 50% = 4

---

### Decision · Program_Chairs · 2026-01-26

Reject